# Neurobehavioral mechanisms of fear and anxiety in multiple sclerosis

Lil Meyer-Arndt[1,2,3,4,5], Rebekka Rust[1,2,3,6], Judith Bellmann-Strobl [1,2,3], Tanja Schmitz-Hübsch[3], Lajos Marko[1,2,7], Sofia Forslund [1,2,7,8], Michael Scheel[3,9], Stefan M. Gold [10,11,12], Stefan Hetzer[13], Friedemann Paul[1,2,3] & Martin Weygandt [1,2] ✉

## Abstract

**Background** Anxiety is a common yet often underdiagnosed and undertreated comorbidity in multiple sclerosis (MS). While altered fear processing is a hallmark of anxiety in other populations, its neurobehavioral mechanisms in MS remain poorly understood. This study investigates the extent to which neurobehavioral mechanisms of fear generalization contribute to anxiety in MS.

**Methods** We recruited 18 persons with MS (PwMS) and anxiety, 36 PwMS without anxiety, and 23 healthy persons (HPs). Participants completed a functional MRI (fMRI) fear generalization task to assess fear processing and diffusion-weighted MRI for graph-based structural connectome analyses.

**Results** Consistent with findings in non-MS anxiety populations, PwMS with anxiety exhibit fear overgeneralization, perceiving non-threating stimuli as threatening. A machine learning model trained on HPs in a multivariate pattern analysis (MVPA) cross-decoding approach accurately predicts behavioral fear generalization in both MS groups using whole-brain fMRI fear response patterns. Regional fMRI prediction and graph-based structural connectivity analyses reveal that fear response activity and structural network integrity of partially overlapping areas, such as hippocampus (for fear stimulus comparison) and anterior insula (for fear excitation), are crucial for MS fear generalization. Reduced network integrity in such regions is a direct indicator of MS anxiety.

**Conclusions** Our findings demonstrate that MS anxiety is substantially characterized by fear overgeneralization. The fact that a machine learning model trained to associate fMRI fear response patterns with fear ratings in HPs predicts fear ratings from fMRI data across MS groups using an MVPA cross-decoding approach suggests that generic fear processing mechanisms substantially contribute to anxiety in MS.

## Plain language summary

Multiple sclerosis (MS) is a chronic autoimmune disease that affects the brain and spinal cord, leading to muscle weakness, tiredness, and cognitive difficulties. Anxiety is common in people with MS, but the brain mechanisms behind it are not fully understood. This study imaged the brain of people with MS responding to learned signals of fear/threat. People with MS and anxiety tended to respond to non-threatening situations as if they were threatening. A computational model was generated based on brain activity from healthy individuals that predicted this pattern in people with MS, suggesting similar fear processing occurs in healthy individuals and people with MS. These findings could help clarify how anxiety in MS may involve general brain mechanisms of threat processing, which could support improved detection and treatment in the future.

Multiple Sclerosis (MS) is frequently accompanied by anxiety[1], with a higher lifetime prevalence of anxiety disorders (ADs) in persons with MS (PwMS; 35.6%) than the general population (29.6%; e.g.,[2]). While anxiety can negatively impact neurocognitive function in MS[3], reduce quality of life[4], and may even be a prodrome of MS[5], it is often underdiagnosed and undertreated[6].

The mechanisms underlying anxiety in MS remain poorly understood, and there is ongoing debate as to whether it rather follows a cognitive reaction to MS progression or whether it is actively promoted by MS-driven pathology (such as degeneration of neural fear processing regions or inflammation and subsequent demyelination of anxiety-related white matter [WM] pathways; refs. 7–9). Specific components of anxiety, like altered fear processing—a key feature of ADs[10]—can be effectively studied using fear conditioning tasks, as shown in studies of anxiety in the general population and psychiatric patients ([11] provides an overview); however, these have rarely been investigated in MS. In basic fear conditioning, a neutral stimulus is paired with an aversive unconditioned stimulus (US; e.g., an electric shock) and becomes a conditioned threat stimulus (CS+) that elicits fear on its own after repeated couplings. Another stimulus, never coupled with the US, becomes a safety cue (CS−). A more complex task with high real-world relevance is the generalization gradient paradigm, which demonstrates how individuals generalize fear from aversive to non-aversive stimuli. In this task, stimuli are selected from a perceptual continuum

(e.g., comprising a large ring as CS+, a small ring as CS−, and rings of intermediate diameter also never coupled to the US but perceptually bridging the gap between the CS+ and the CS− as generalization stimuli; GS). Individuals with panic disorders[12], generalized anxiety disorder[13], and posttraumatic stress disorder[14] exhibit less steep declines in fear responses to GS that are increasingly dissimilar from the CS+ compared to HPs. This results in flatter gradients, indicating fear overgeneralization. Across neuroscience and psychiatric studies[15,16], hippocampal functioning appears pivotal importance for fear generalization, as it mediates the comparison of stimulus representation to the CS+; if this comparison is inaccurate or biased, overgeneralization is promoted.

Thus, motivated by the importance of anxiety in MS, the scarcity of research on its mechanisms, and insights gained from fear conditioning tasks into the pathophysiology of anxiety in the general population and psychiatric cohorts, we employed a validated functional MRI (fMRI) fear generalization gradient task[17] combined with diffusion-weighted imaging (DWI) MRI to study fear generalization at behavioral, functional and structural level in MS.

18 PwMS with anxiety (PwMSA), 36 PwMS without anxiety (PwMSNA), and 23 healthy persons (HPs) were studied. We computed behavioral generalization based on shock-risk ratings to test the hypothesis that PwMSA exhibit fear overgeneralization similar to individuals with anxiety but without MS, as observed in non-MS studies (e.g.,[11]). We tested the hypothesis that fear generalization recruits overlapping neural processing systems across groups by employing a multivariate pattern analysis (MVPA) cross-decoding approach[18], evaluating whether a machine learning (ML) algorithm, trained on rating and fMRI fear response data from HPs, could predict risk ratings of PwMSA and PwMSNA based on their fMRI data. Finally, we related patients' behavioral fear generalization to graph-based structural brain network parameters (e.g.,[19]) derived from DWI, to test our hypothesis that network integrity of generic fear generalization areas reflects behavioral generalization in MS.

These analyses show that, similar to individuals with anxiety but without MS, PwMSA tend to overgeneralize fear, responding to non-threatening cues as if they pose a threat. A machine learning model trained on HPs successfully predicts behavioral fear generalization in both MS groups using whole-brain fMRI patterns. Further analysis of regional fMRI data and structural brain networks indicates that increased activity and network integrity in partially overlapping brain regions, such as hippocampus involved in comparing fear stimuli and anterior insula linked to fear reactivity, corresponds to stronger fear generalization in PwMS. Moreover, lower structural network integrity in these regions corresponds with greater anxiety symptoms in PwMS.

## Materials and methods
### Participants
58 PwMS and 33 HPs were recruited through Charité outpatient clinics and advertisements. Participants underwent clinical assessments and MRI scans within a two-week period. Inclusion criteria for PwMS included age 18–65 years, relapsing-remitting or secondary progressive MS according to the revised McDonald criteria[20], stable or no immunomodulatory treatment for the past six months, and the mental and physical capability for study participation. Exclusion criteria encompassed other neurological and immune-mediated disorders or psychiatric disorders other than ADs and depressive disorders, relapses or steroid treatment during the last four weeks, and MRI contraindications. Where applicable, inclusion and exclusion criteria for HPs were identical. Applying these criteria yielded 54 PwMS. Twenty-nine HPs met the criteria, but only 23 fulfilled rating criteria for behavioral task ratings and are thus referenced throughout, as HPs were only included in analyses involving these ratings.

### Inclusion and ethics
All participants provided informed consent in accordance with the Declaration of Helsinki. The study protocol was approved by the ethics committee of Charité—Universitätsmedizin Berlin (EA1/209/19). The study population included adult participants of diverse gender identities and age groups, and efforts were made to ensure inclusivity and representativeness. No vulnerable populations were targeted or excluded. All methods were carried out in accordance with relevant guidelines and regulations.

### Clinical assessment
The study physician assessed clinical disability using the Expanded Disability Status Scale (EDSS[21]) and evaluated the presence of neurological, psychiatric, and immune-mediated diseases, occurrence of relapses, use of steroids, immunomodulators, or antidepressants, and MRI-related contraindications. Anxiety was assessed using the trait scale of the German version of the STAI (STAI-T; ref. 22), which measures the stable propensity to experience anxiety. A cutoff of ≥ 41 points ([23]) classified anxiety, resulting in 18 PwMSA and 36 PwMSNA. Information processing capacities were assessed by measuring the average time required for risk rating during the fMRI task (see below). Self-report data were collected with the Beck Depression Inventory II (BDI-II[24]) and the Modified Fatigue Impact Scale (MFIS[25]).

### fMRI fear generalization task
We implemented an fMRI task based on the paradigm developed by Lissek et al.[17] to study neurobehavioral fear generalization in MS. The task comprised three stages: pre-acquisition (assessing baseline responses to stimuli in a neutral, unconditioned state), acquisition (during which participants learned the association between US – and CS+, and the absence of US - CS-associations), and generalization (assessing responses to stimuli following conditioning). Throughout all stages, participants were instructed to rate the perceived risk of shock as quickly as possible upon presentation of the rating cue. Fig. 1 outlines key task features. Additional details, such as shock application and the calibration procedure used to individuals adjust shock intensity (i.e., US) prior to MRI, are provided in the Supplement (see "fMRI fear generalization task" in "Supplementary methods").

### MRI sequences
All MR images were acquired with the same 3 Tesla whole-body tomograph (Magnetom Prisma, Siemens, Erlangen, Germany) and 64-channel head coil. Acquisition of anatomical MRI scans comprised a saggital T1-weighted and FLAIR sequence. Functional scans were acquired using a T2*-weighted multi-band Echo-Planar-Imaging (EPI) Blood-Oxygen-Level-Dependent (BOLD) sequence from the Human Connectome Project26. Additionally, two spin-echo EPI reference volumes with opposing phase encoding directions were acquired prior to the first fMRI run with matching readout and geometry for conducting a distortion correction of fMRI scans. DWI data required were acquired with a multi-shell DWI MRI sequence from the Human Connectome Project[26]. Also, for DWI distortion correction, we acquired pairs of spin-echo EPI reference volumes with opposing phase-encoding directions with matching readout and geometry. For details, see "MRI sequences" in "Supplementary methods" in the Supplement.

### Processing of anatomical MRI scans
This comprised a manual lesion mapping using FLAIR scans, a combined spatial normalization and segmentation of T1-weighted images to the anatomical standard space defined by the Montreal Neuological Institute (MNI[27];) with SPM12 (Wellcome Trust Centre for Neuroimaging, Institute of Neurology, University College London, London, UK), and a determination of a gray matter (GM) group mask covering the entire brain (see "Processing of anatomical MRI scans" in "Supplementary methods" in the Supplement).

### fMRI preprocessing and brain activity modeling
We linearly coregistered the EPI reference volume with posterior-to-anterior encoding direction and the fMRI scans of each run to the space of the EPI reference volume with anterior-to-posterior encoding direction

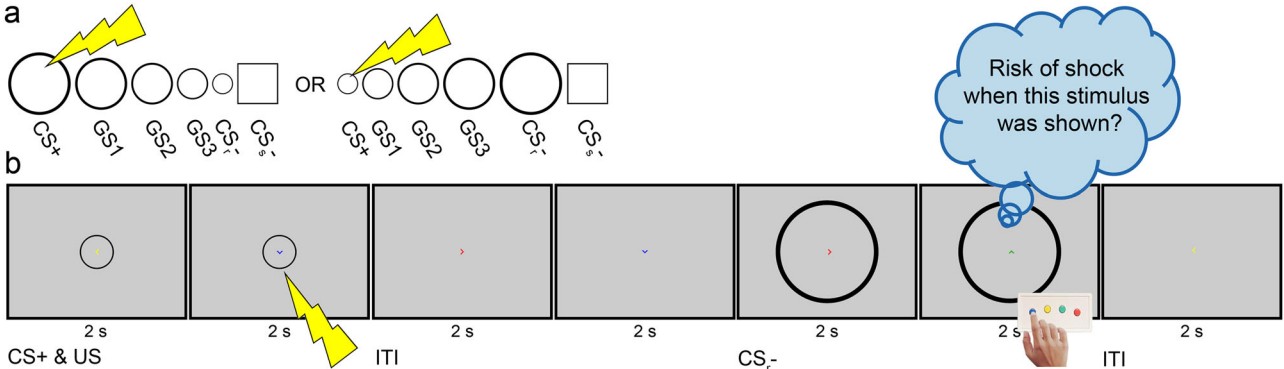

**Fig. 1 | fMRI fear generalization task.** The fMRI task comprises three consecutive stages: Pre-acquisition (two runs), acquisition (one run), and generalization (two runs). **a** depicts the stimuli displayed to the participants: five ring-shaped stimuli (i.e., the CS+, a ring-shaped CS− [CS$_r$−], and three GS) and a square-shaped CS− (CS$_s$−). During pre-acquisition, all six stimuli were shown in eight trials per run each, but none was ever paired with the shock. During acquisition, the CS+, CS$_r$− and CS$_s$− were shown in 15 trials each, and the CS+ co-terminated with an electrical shock (US) in twelve trials. The generalization stage was identical to pre-acquisition except that four trials were included additionally in which the CS+ co-terminated with a shock to prevent fear extinction. Prior to task onset, the participants were instructed that they might learn to predict the occurrence of a shock if they attend to the depicted stimuli. **b** illustrates two trials of a generalization run. Across stages, trials were composed of time bins with two seconds duration. Each trial started with the presentation of a CS or GS (two bins duration) and was followed by an ITI of one to three bins (average: two bins) duration. Additionally, one of four different arrowheads (a green one pointing up, yellow - left, red - right and blue - down) was shown in the middle of the screen during a bin. The bin-wise structure served two purposes. First, the appearance of a green arrowhead signaled the participant to rate the perceived risk of shock (ranging from minimal to moderate to maximal) for the currently depicted ring- or square-shaped stimulus as quickly as possible using an MRI-compatible response box. Second, the participants were instructed to focus the center of the arrowheads to minimize their head motion. For further details, see Supplement. Abbreviations: CS+ conditioned stimulus, CS− safety cue, fMRI functional MRI, GS generalization stimulus, ITI inter-trial-interval, US unconditioned stimulus.

with SPM12. Afterwards, we conducted a distortion correction of the fMRI scans per run with the FSL-routine top-up[28,29] based on the EPI reference volumes. Next, the fMRI scans of a run were realigned to correct for head motion. After averaging the realigned scans, the mean image was linearly coregistered to the T1-weighted scan, and the parameters determined were used to linearly coregister the full series of fMRI scans to the T1-weighted scan. By then, applying the deformation field computed for the T1-weighted image during the combined normalization and segmentation of T1-weighted scans to the coregistered fMRI scans, these were mapped to MNI space. Finally, a spatial smoothing (8 mm full-width at half maximum Gaussian kernel) and a temporal high-pass filter (128 s cut-off) were performed.

Following[17], modeling of brain activity voxel timeseries during the two generalization runs was then conducted based on the preprocessed scans per participant in a run-wise fashion with a General Linear Model (GLM) and a design matrix including eight regressors reflecting the timing of task components, five of which were regressors of interest—one for each ring-shaped stimulus (i.e., the CS+, GS1 – 3, and the CS$_r$−). Additionally, it contained a regressor for the CS$_s$−, one for trials where the CS+ was paired with the US, and one for the button presses performed for risk rating. Besides these regressors reflecting the timing of task components, another six regressors (derived from the realignment of fMRI scans during preprocessing) were included in the design matrix, reflecting the participants' head motion. Except for the regressor coding button presses, boxcar regressors coding ones for the two-time bins per trial presenting CS or GS and zeros for the ITI time bins were determined first for the eight regressors reflecting task components. For button presses, the boxcar regressor coded ones for the period from the onset of the rating bin to the time of the button press and zeros for the remaining time. After these boxcar regressors were defined, they were convolved with the hemodynamic response function to account for the temporal characteristics of the BOLD response. The full design matrix of all fourteen regressors was then analyzed with the GLM to compute the neural responsivity to each regressor. Voxel-wise regression coefficient maps for each of the five regressors of interest per participant and generalization run were entered into the fMRI analyses as training and test patterns for a Support Vector Regression (SVR) model. Based on the

inclusion criteria defined for risk ratings and the availability of fMRI data, all 230 patterns were available for the 23 HPs, 130 for the 13 PwMSA, and finally 305 for the 31 PwMSNA.

### Preprocessing of DWI scans, tractography, and graph-based connectivity modeling

We first linearly coregistered the EPI reference volume (posterior to anterior encoding direction) as well as the DWI scans to the space of the EPI reference volume (anterior to posterior encoding direction). Distortion correction of the DWI scans was then performed using FSL based on the two EPI reference volumes. Next, we corrected for head motion and eddy current-induced artefacts using FSL.

A probabilistic Anatomically Constrained Tractography (ACT; ref. 30) was performed using algorithms from Mrtrix3[31], FSL, and SPM12 on multi-shell multi-tissue DWI scans. Specifically, we estimated diffusion basis functions from the individuals' multi-shell, multi-tissue DWI scans and computed fiber orientation densities, which were deconvolved with the basic functions using the constrained spherical deconvolution approach in Mrtrix3. We then created a GM/WM boundary map for each participant from their MPRAGE scans to define physiologically plausible regions as start and stop points for the streamline tracing algorithm in Mrtrix3. This boundary (segmented GM maps from the previous step) was coregistered to the participants' DWI scans, and ACT was performed. In this process, streamline length was limited to 250 mm, the fiber orientation distribution cutoff for streamline termination was set to 0.06, and 10 million streamlines were computed per participant.

We subsequently performed inverse normalization of the Neuro-morphometrics brain atlas (defined in MNI space) to coregister it to the same space as the participants' DWI scans using SPM12. The structural connectivity matrix, reflecting the number of streamlines connecting pairs of regions in the coregistered Neuromorphometrics brain atlas, was computed using Mrtrix3.

Finally, these matrices were entered into the Brain Connectivity Toolbox[32] to compute, individually for each region in the Neuromorphometrics brain atlas and each participant, three regional connectivity measures relevant across various brain disorders, including MS (e.g.,[19]). These measures included: regional degree (indicating the number of connections

between a region and its neighboring regions), betweenness centrality (quantifying how often a region lies on the shortest paths between other region pairs), and clustering coefficient (reflecting the extent to which neighboring regions of a given region are also interconnected).

### Statistical analysis

**Behavioral fear generalization.** We tested our hypothesis that PwMSA exhibit behavioral overgeneralization of fear based on the risk ratings for all ring-shaped stimuli acquired across the two generalization runs. Three quality assurance steps were conducted beforehand: We (i) included only those participants fulfilling rating criteria, ensuring they fully engaged with and understood the task. Specifically, we constrained the analysis to participants who showed variation in their ratings (i.e., did not rate the same risk for all stimuli), provided at least half of all 40 possible ratings for ring-shaped stimuli across both generalization runs, and rated a risk probability increasing from the $CS_r-$ to the CS+. This resulted in a selection of 13 PwMSA, 31 PwMSNA, and 23 HPs. Next, we confirmed (ii) that the proportion of participants exposed to the smallest or largest ring as the CS+ was similar across groups to exclude effects of physical stimulus properties and (iii) that the task successfully induced fear by evaluating behavioral fear responses during pre-acquisition and acquisition (see "Statistical analysis" in the Supplement and Supplementary Fig. 1 for details on steps [ii] and [iii]).

To mathematically characterize behavioral fear generalization gradients for each participant individually (see "Statistical analysis" in "Supplementary methods" in the Supplement for details), we modeled their risk ratings with logistic regression, a key method for the analysis of choice data in judgment and decision-neuroscience[33] and computed the so-called point of indifference (PI) as subject-specific fear generalization measure. The PI corresponds to the point in a parametric stimulus space, for which the preference for all options is identical[33]. Thus, here, the PI corresponds to the point on the stimulus continuum ranging from the $CS_r-$ to the CS+, for which the logistic model computed a rated risk probability of 0.5.

Finally, we tested group differences in the PIs using multiple regression. A dichotomous group vector served as covariate of interest (CI). The covariates of no interest (CNI) for the comparison PwMSA vs. PwMSNA included: Average rating time (to control for information processing speed), log-transformed volume of hyperintense lesions in FLAIR scans, disease duration, disease type (i.e., relapsing remitting vs. secondary progressive MS), age, sex, use of interferon β (y/n; due to their ability to induce depression and anxiety[34];) and antidepressants (y/n). For the group comparisons, including HPs, average rating time, sex, age, and the log-transformed lesion volume served as CNI. Permutation testing with 20,000 iterations of the CI vector and a one-sided significance threshold of $\alpha = 0.05$ (assuming generalization in PwMSA > PwMSNA > HP) was employed for inference. Cohen's $f^2$ is reported as effect size measure ($f^2 \geq 0.02$, 0.15, and 0.35 correspond to a weak, moderate, and strong effect). See "Statistical analysis" in "Supplementary methods" in the Supplement for an analysis additionally modeling fatigue and depression as CNI (Supplementary Fig. 2).

**Neural substrates of behavioral fear generalization.** We employed an MVPA cross-decoding ML approach[18,35–37] to test our hypothesis that fear generalization recruits overlapping neural processing systems in HPs and PwMS with and without anxiety. Briefly, MVPA studies assume that patterns of neural activity reflect the structure of mental representations and use ML algorithms to determine whether these patterns vary systematically along a continuous response variable, such as fear ratings (e.g.,[18]). The principle behind cross-decoding is that 'if an ML algorithm trained on patterns from one context performs well when tested on patterns acquired in another, then the representations of the variable of interest are similar across both contexts'[38].

Here, an SVR algorithm implemented in Matlab (MathWorks, Natick, Massachusetts, USA) during training learned to associate 230 fMRI fear response (i.e., regression coefficient) patterns of persons

neither affected by MS nor anxiety (i.e., of HPs) with the associated perceived risk. Consistent with the analysis of behavioral fear generalization, perceived risk was computed by the subject-specific logistic regression models based on the ratings for the different ring-shaped stimuli (i.e., the labels shown as orange dots in Fig. 3). In testing, we used the HP-derived SVR model for predicting the true perceived risk (modeled by logistic regresion) based on the 130 voxel patterns derived from PwMSA and the 305 voxel patterns from PwMSNA. The correlation between true perceived risk and that predicted by SVR served as accuracy measure. Default SVR hyperparameters predefined by MATLAB were used. Permutation testing (20,000 permutations of training labels) was used for statistical inference in one-sided tests. Bootstrapping tested the accuracies' robustness against (i.e., independence of) variations in the specific distribution of patterns and labels (1000 resamplings of the HPs' training data), which was expressed in terms of the 95% accuracy confidence intervals ($CI_{95\%}$) computed across the 1000 accuracies obtained per patient group. Potential effects of demographic and disease-related nuisance factors on accuracy were tested in the Supplement (see "Statistical analysis" in "Supplementary methods") by repeating this analysis with fMRI patterns adjusted for CNI.

To evaluate the contribution of individual brain regions in both MS groups, we repeated the whole-brain GM analysis based on activity of coordinates located in the intersection of individual GM regions defined by the brain atlas and the GM group mask per atlas region. 10,000 permutations per region were performed for inference (one-sided tests). The significance threshold was adjusted for family-wise error (FWE) with the Bonferroni method (i.e., by dividing the α-level for a single test [0.05] by the number of regions [i.e., 120], yielding $4.2 \cdot 10^{-4}$). To evaluate regional neural substrates of altered fear processing in PwMSA, we tested for regions showing group differences in accuracy for PwMSA vs. PwMNSA using a Fisher Z-test[39] and an FWE-corrected threshold.

**Structural brain connectivity, behavioral fear generalization, and anxiety in MS.** We tested our hypothesis that network integrity of generic fear generalization areas described in basic and neuropsychiatric research (e.g.,[16,17,]) reflects behavioral generalization in MS, based on data of the 12 PwMSA and 30 PwMSNA fulfilling the rating criteria and having a complete DWI data set.

The three connectivity indices (regional degree, betweenness centrality, and clustering coefficient), computed separately for each region in the Neuromorphometrics brain atlas, were used as CI in linear regression analyses modeling patients' PIs. These analyses were conducted separately for each of the 142 regions and each connectivity measure across all 42 patients. CNI was as in analyses of group differences in behavioral generalization for PwMSA vs. PwMSNA. Permutation testing (20,000 permutations) was applied for two-sided significance testing, with Bonferroni correction for FWE. In a post-hoc analysis, connectivity parameters found significant above were tested for their ability to directly characterize MS-related anxiety using data from all 17 PwMSA and 35 PwMSNA with available DWI scans. Here, a dichotomous group vector served as the CI, and CNI was as in the prior analysis, except information processing speed was excluded as rating data were only available for 42 patients, and we aimed to utilize the full DWI dataset of 52 PwMS. To assess the potential influence of information processing speed, we also conducted a complementary analysis constrained to the 42 patients with available rating data, including it as a CNI. Permutation testing (20,000 permutations) was used for inference, with Bonferroni correction for FWE in one-sided tests. The Supplement (see "Statistical analysis" in "Supplementary methods") additionally includes analyses incorporating fatigue and depression as CNI.

## Results
### Clinical and demographic participant characteristics
Forty MS patients received immunomodulatory treatment. The MS groups did not differ in any of these treatments. Four patients received

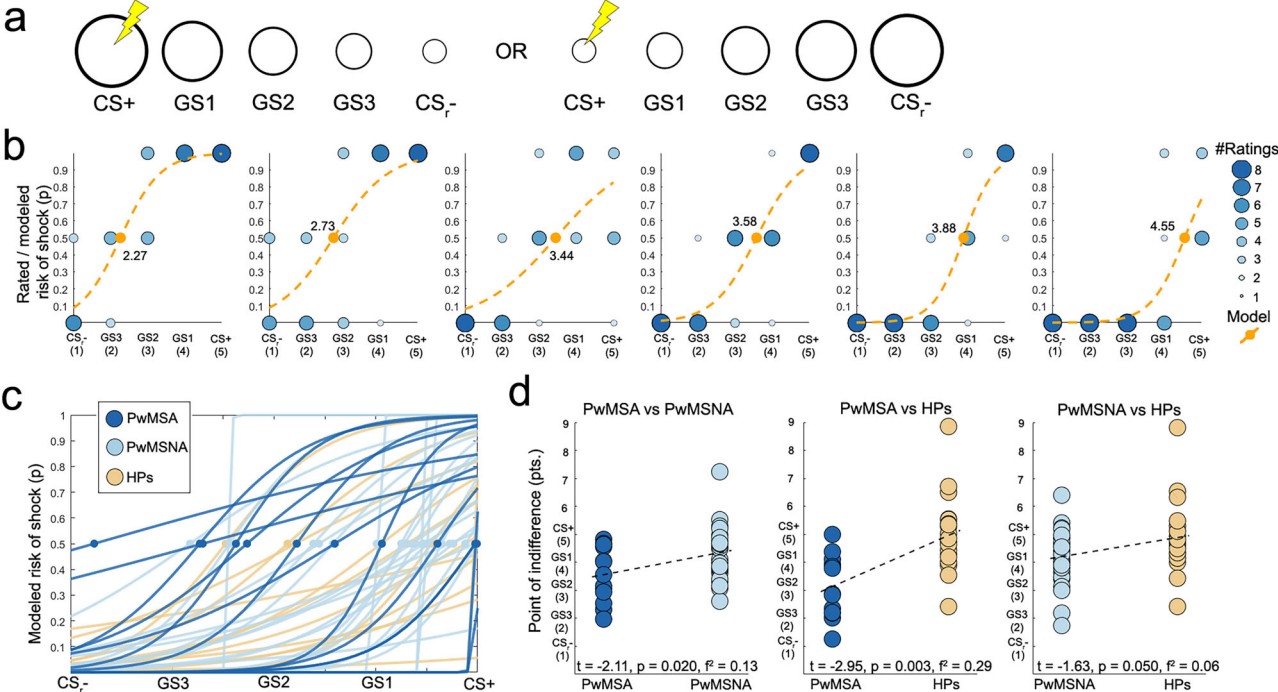

**Fig. 2 | Behavioral fear generalization. a** depicts the ring-shaped stimuli presented. **b** shows behavioral risk rating data for six exemplary participants (from left to right: for two PwMSA, two PwMSNA, and two HPs) as blue dots. The diameter and color intensity of blue dots reflect the frequency with which each of the five ring-shaped stimuli was rated as minimal risk (i.e., 0), moderate risk (0.5), or maximal risk (1) across the two generalization runs. The dashed orange line depicts the logistic regression model used to mathematically characterize the risk ratings or generalization gradients, respectively, the orange dot shows the PI derived from this model. Smaller PI values (i.e., located closer to the $CS_r-$) indicate more pronounced fear

generalization. **c** depicts the logistic regression models/the PI of all 67 participants (PwMSA $n = 13$, PwMSNA $n = 31$, HPs $n = 23$) separated by group. Finally, **d** illustrates the results of multiple regression analyses testing the differences in PI between all three pairs of groups. The depicted PI values are corrected for effects of CNI (and might thus slightly differ across group comparisons). Abbreviations: CNI covariates of no interest, $CS_r-$ ring-shaped safety cue, HPs healthy persons, PI point of indifference, PwMSA persons with multiple sclerosis and anxiety, PwMSNA persons with multiple sclerosis without anxiety.

antidepressants. All groups were comparable in terms of demographic parameters, and both MS groups were comparable in terms of information processing capacities, GM fraction, T2-weighted lesion volume, disease duration, relapsing or secondary progressive disease type, and clinical disability. In line with the high prevalence and frequent comorbidity of anxiety, depression, and fatigue in MS[40], PwMSA had higher scores on both parameters compared to the other two groups, while PwMSNA had higher scores than HPs. See Supplementary Data 1 for details.

### Behavioral fear generalization

PwMSA exhibit stronger fear generalization (i.e., smaller PI scores) than PwMSNA ($t = -2.11$, $p = 0.020$, $f^2 = 0.13$) and HPs ($t = -2.95$, $p = 0.003$, $f^2 = 0.29$). PwMSNA demonstrated stronger generalization than HPs at a relaxed significance level of $\alpha = 0.1$ ($t = -1.63$, $p = 0.050$, $f^2 = 0.06$; Fig. 2).

### Neural substrates of behavioral fear generalization

An HP-derived SVR model predicted perceived risk ratings with significant accuracy in both patient groups based on fMRI response patterns. The accuracy was highly robust to variations in the distribution of patterns and labels (PwMSA: $r = 0.59$ [i.e., 35% of the variance in the ratings were explained], $p < 5 \cdot 10^{-5}$, $CI_{95\%} = [0.38\ 0.67]$; PwMSNA: $r = 0.58$ [34% of the variance], $p < 5 \cdot 10^{-5}$, $CI_{95\%} = [0.43\ 0.62]$; see Fig. 3A). A supplementary analysis (see "Statistical analysis" in "Supplementary methods" in the Supplement), which adjusted fMRI patterns for demographic and disease-related CNI, demonstrated that the accuracy remained highly significant. Fig. 4 provides an overview of the contributions of individual atlas regions to risk prediction for both MS groups. Notably, accuracy did not differ between the two groups for any of these regions.

### Structural brain connectivity, behavioral fear generalization, and anxiety in MS

We found significant (positive) associations between the PIs and the clustering coefficient of the left inferior temporal gyrus ($t = 4.40$, $p_{FWE} = 0.012$, $f^2 = 0.61$), and left hippocampus ($t = 4.00$, $p_{FWE} = 0.043$, $f^2 = 0.50$). A post-hoc analysis showed that the clustering coefficient of the left inferior temporal gyrus was significantly lower in PwMSA compared to PwMSNA ($t = -2.44$, $p_{FWE} = 0.016$, $f^2 = 0.14$; Fig. 5). This effect remained significant when information processing speed was included as a CNI ($t = -2.28$, $p_{FWE} = 0.028$, $f^2 = 0.16$. Incorporating fatigue and depression as additional CNI demonstrated that the observed effects were comparably robust to both factors (Supplementary Fig. 3).

### Discussion

Anxiety is a prevalent and important comorbidity in MS[1,4,6], yet the underlying neurobehavioral mechanisms remain poorly understood. Using an fMRI fear generalization task and DWI, our study shows that fear generalization in MS relies to a substantial degree on generic mechanisms.

We found that PwMSA overgeneralize fear behaviorally, exhibiting less steep decline in fear responses to stimuli increasingly dissimilar to CS+ than PwMSNA and HPs. This aligns with findings in various ADs[12–14]. Our results remained significant for PwMSA vs. HPs and marginally significant for PwMSA vs. PwMSNA ($p = 0.068$) after controlling for depression and fatigue (Supplementary Fig. 2), suggesting specificity to anxiety. Our work extends Ellwardt et al.'s study[41], which first used a fear conditioning paradigm in MS but focused on electroencephalographic measures characterizing the entire measurement period rather than key parameters for individual CS and compared patients with normal anxiety levels to HPs and thus did not study anxiety in MS. Additionally, the application of

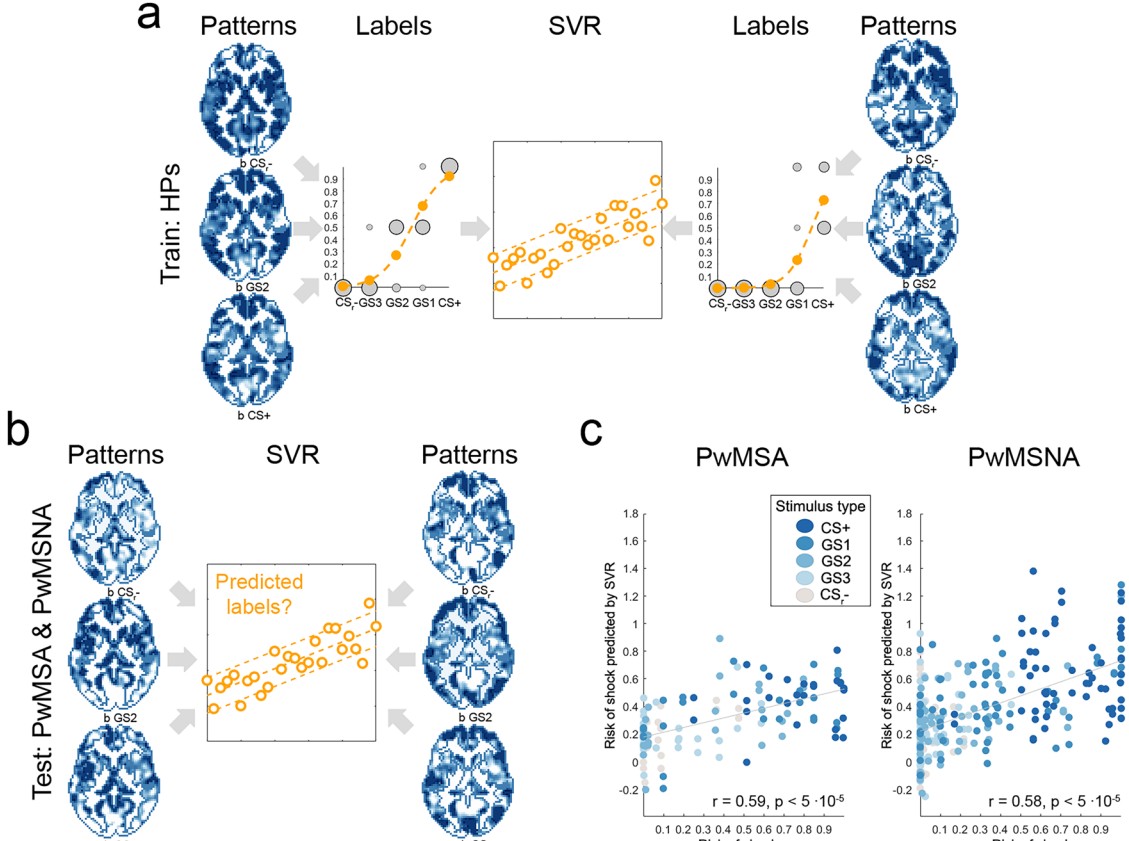

**Fig. 3 | Predicting perceived risk of shock based on brain activity distributed across the entire GM. a** Training phase of the SVR-based cross-decoding model using data from two randomly selected HPs and one of two available generalization runs per HP. The model was exclusively trained on HPs' data, using voxel-wise regression coefficient maps **b** computed for each of the five ring stimuli (i.e., CS−, GS3, GS2, GS1, CS+) as described in the Methods section "fMRI preprocessing and brain activity modeling". For space reasons, only three of the five maps ($b_{CSr}$, $b_{GS2}$, and $b_{CS+}$) are depicted here. The corresponding "Label" graphs in 3 A depict the

perceived risk of shock for each stimulus, modeled via logistic regression, with values indicated by orange dots. **b** Illustration of the testing procedure. **c** Prediction accuracy for the patterns of PwMSA ($n = 13$) and PwMSNA ($n = 31$). Abbreviations: CS+ conditioned stimulus, CS− safety cue, GM gray matter, GS1 generalization stimulus 1, GS2 generalization stimulus 2, GS3 generalization stimulus 3, HPs healthy persons, PwMSA persons with multiple sclerosis and anxiety, PwMSNA persons with multiple sclerosis without anxiety, SVR support vector regresion.

---

transcranial magnetic stimulation during their task complicated the results' interpretation, as observed effects could reflect fear processing or stimulation.

By showing that an SVR model trained to associate fMRI patterns and fear ratings in HPs predicted fear rating in PwMSNA (explaining 34% of the variance) in an MVPA cross-decoding analysis, the study demonstrates that generic neural mechanisms are important drivers of fear ratings in MS. Moreover, the model's ability to generalize to PwMSA (35% explained variance) indicates that a common fear processing system accounts for the full range of generalization—from low levels in HPs to overgeneralization in PwMSA. These findings (of more than one third of the variance in fear ratings in both patient groups explained by generic fear mechanisms) suggest that anxiety in MS may arise from a combination of generic and MS-specific mechanisms. Importantly, the functional nature of these processes, amenable to modulation via neuropsychiatric interventions[42], points to possible avenues for treating anxiety in MS.

MVPA analyses conducted per atlas region separately for both patient groups revealed significant accuracy in (amongst others) anterior insula, hippocampus, and lateral orbitofrontal cortex, particularly in PwMSNA. These regions align with a generic fear generalization model[15], in which hippocampal stimulus matching determines the deployment of excitatory or inhibitory fear processes. Hippocampus compares incoming stimuli (e.g., the GS1) to threat-related memories (e.g., the CS+). High similarity activates a representation of the threat-related stimulus (i.e., pattern completion), engaging threat excitation areas (including the anterior insula).

Conversely, low similarity (i.e., pattern separation) engages the ventromedial prefrontal cortex (vmPFC) to inhibit fear and signal safety. While vmPFC was not identified, lateral OFC, an area able to impair vmPFC safety signaling[43], was, which potentially provides an explanation for this absence. Please note that the discrepancy between regional prediction accuracies (few significant regions, max $r = 0.44$) and whole-GM accuracies ($r_{PwMSA} = 0.59$, $r_{PwMSNA} = 0.58$) is compatible with a similar discrepancy found by Liu et al.[44]. Who could explain it with the distributed nature of fear processing only reflected by whole-GM prediction. The lack of significant regional accuracy differences between PwMSA and PwMSNA supports the idea that a single neural system is underlying fear generalization across the full spectrum of (over-)generalization.

Finally, we examined how structural connectivity of local networks reflects behavioral generalization in MS. This is essential due to frequent white matter (WM) pathology-induced disconnections in MS and the distributed nature of fear processing, which is naturally vulnerable to such disconnections[44]. Aligning with Lissek's model[15], our analysis revealed that stronger connectedness among network neighbors (i.e., higher clustering coefficients) of left hippocampus and inferior temporal gyrus correlated with less fear overgeneralization. This underlines the importance of intact cross-regional information flow in these neighborhoods as a protective mechanism against fear overgeneralization. To assess whether these differences characterize MS-related anxiety directly, we conducted a post-hoc analysis comparing the clustering coefficients of these regions between PwMSA and PwMSNA. A significant difference

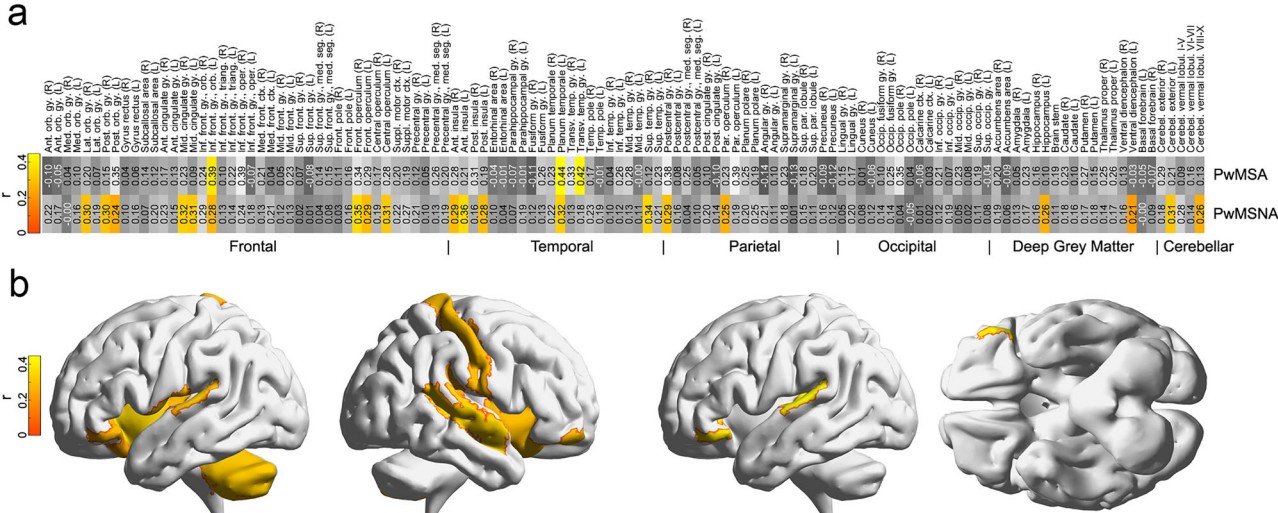

**Fig. 4 | Predicting the perceived risk of shock based on activity of individual brain regions.** The heatmap in **a** depicts the accuracies (i.e., Pearson correlation coefficients r between true and predicted labels) separately for the individual atlas regions and patient groups. Elements shown in yellow-orange correspond to areas significant according to permutation testing and the FWE-corrected significance threshold. Please note that due to the resampling procedure in permutation testing, an extreme correlation must not always be less probable than a slightly less extreme one. Further, the assignment of regions to lobes was made exclusivley to ease the readability of the graph and is not necessarily anatomically optimal. **b** illustrates the regions for which significant decoding accuracies were obtained for: The two renderings on the left half depict those for PwMSNA ($n = 31$), the other two on the right side those for PwMSA ($n = 13$). Abbreviations: FWE, family-wise error; PwMSA, persons with multiple sclerosis and anxiety; PwMSNA, persons with multiple sclerosis without anxiety.

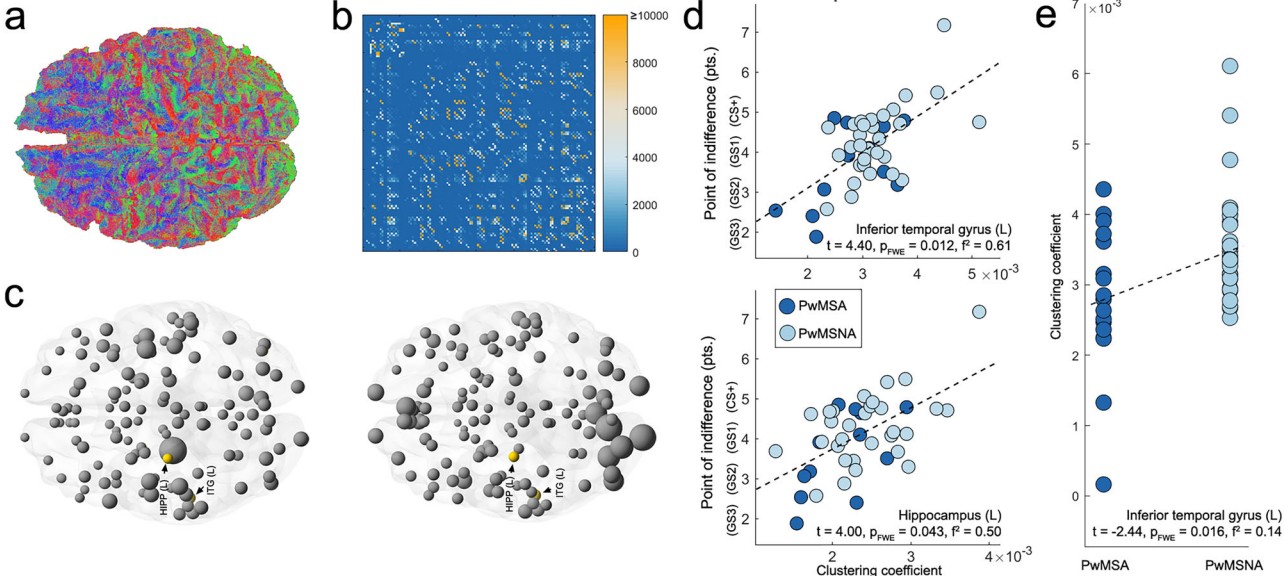

**Fig. 5 | Structural brain connectivity and behavioral fear generalization in MS.** To demonstrate the outcome of the Anatomically Constrained Tractography conducted for the whole brain, **a** shows the tractography generated by Mrtrix3 for an arbitrarily selected person. **b** depicts an exemplary structural connectivity matrix characterized by the number of streamlines connecting pairs of regions that resulted from mapping the Neuromorphometrics brain atlas to that tractography. **c** Shows the clustering coefficients for the regions in the atlas for two arbitrarily selected MS patients derived from such matrices. The diameter of the spheres depicts the regions' clustering coefficients for each of the two individuals. Labeled regions shown in orange correspond to areas for which a significant positive association between the regional clustering coefficient and the PI was found according to the Bonferroni-corrected significance threshold across all patients in the analysis. **d** Further illustrates these significant associatons (PwMSA $n = 12$, PwMSNA $n = 30$). Finally, (**e**) illustrates the significantly smaller clustering coefficients in PwMSA ($n = 17$) than PwMSNA ($n = 35$) in left inferior temporal gyrus found in the post-hoc analysis. Abbreviations: PI point of indifference, PwMSA persons with multiple sclerosis and anxiety, PwMSNA persons with multiple sclerosis without anxiety.

was found for left inferior temporal gyrus. Adjusting for fatigue and depression suggested that these findings are largely specific to anxiety (Supplementary Fig. 3).

An aspect that should be discussed is the possibility that the identified generic mechanisms—or alterations in regions known to contribute to generic fear processing—might represent risk factors for developing both anxiety and MS, with fear generalization emerging as an expected behavioral concomitant. However, given that these mechanisms and regions have been identified across many studies analyzing HPs (e.g.,[15,16,45]), and considering the prevalence of MS (of e.g., 35 per 100,000 individuals globally in 2020 according to[46]), this possibility does not challenge the notion that these mechanisms are indeed generic. Furthermore, given, for example, the strong negative link

between behavioral fear generalization and clustering coefficient in hippocampus or inferior temporal gyrus found in our structural connectivity analysis—which included only PwMS across a wide range of behavioral generalization and structural connectivity levels—it appears unlikely that e.g., lower connectivity in such regions is a strong risk factor for developing MS.

Our study has some limitations. The number of participants might be considered moderate, but the 54 PwMS included here were pre-calculated based on data from one of our studies (Meyer-Arndt et al.[47]), which investigates fMRI activity differences between PwMS with and without depression during processing of affectively negative and neutral pictures. As we also measured (but did not analyze) STAI-T in Meyer-Arndt et al.[47], we could compute an effect size and thus a sample size for associations between brain activity in the affective picture task and STAI-T as a reference for this fear conditioning study. As pre-calculated, we included 54 PwMS, but this fixed number could not account for multi-paradigm data loss (i.e., insufficient rating behavior and missing DWI scans). Consequently, to the extent that an affective picture task can provide an effect/sample size for a fear conditioning task, one might argue that our study is slightly underpowered. However, first, moderate to strong effects were found across analyses, and the sample size was thus not too small for supporting our hypotheses. Second, on practical grounds, our study can be considered large compared to existing MS fMRI studies using cognitive tasks: according to a recent review by Rocca et al.[48] the 44 PwMS and 23 HPs included in the crucial fMRI analyses exceed the sample sizes used in 81% of the reviewed cognitive studies.

Furthermore, the inclusion of only a single visit could be noted, as a longitudinal design would have allowed exploration of changes in fear generalization/anxiety ratings alongside structural and functional MRI alterations. However, the choice of a cross-sectional design was predefined by the grant module funding the project, which had a three-year duration. This limited timeframe, especially given partial overlap with the COVID-19 pandemic, posed a risk to successful multi-visit participant acquisition.

Another point is the use of STAI-T instead of, e.g., DSM criteria for ADs assessed by structured interviews, as the latter might have provided more fine-grained measures of anxiety. However, we chose STAI-T as it has been shown (i) to be highly sensitive to anxiety across ADs[49], (ii) to reflect neural fear conditioning responsivity[50], and (iii) because the heterogeneity of ADs would have complicated the acquisition of a homogeneous group of PwMSA during the project's three-year funding period.

Further, we focused on predicting risk from responses to ring-shaped CS acquired in generalization runs because evaluating responses to square-shaped CS or fMRI response differences between generalization and pre-acquisiton would have been incompatible with the parametric CS continuum approach. While tractography in MS might be sensitive to lesions[51], we mitigated this by using a modern multi-shell DWI sequence[26], which is considered robust to increased isotropic diffusion in lesions[51].

In conclusion, our study demonstrates that anxiety in MS is characterized by fear overgeneralization. Much of the perceived fear in PwMS can be explained by generic functional fear processing mechanisms, without invoking MS-specific factors.

### Reporting summary

Further information on research design is available in the Nature Portfolio Reporting Summary linked to this article.

### Data availability

The data supporting the findings of this study are subject to a confidentiality agreement, and participants did not provide consent for public release of their raw data. Therefore, only highly processed or aggregated data are available through the Zenodo data and code repository ([52]; https://zenodo.org/records/16218570). Available data include the clinical and demographic participant characteristics, specifically the information processing speed metrics depicted in Supplementary Data 1; the source data for Supplementary Data 1 are provided in ref. 52 (https://zenodo.org/records/16218570). For the behavioral fear generalization analysis, the data

underlying Fig. 2b, c are provided in ref. 52 (https://zenodo.org/records/16218570). Regarding the neural substrates of behavioral fear generalization, the repository includes raw behavioral data and run-specific SPM fMRI regression coefficient maps resulting from "fMRI preprocessing and brain activity modeling". These files allow full replication of the statistical parameters depicted in Figs. 3 and 4, and the source data are provided in ref. 52 (https://zenodo.org/records/16218570). For the analysis of structural brain connectivity, behavioral fear generalization, and anxiety in MS, structural connectivity matrices are available in ref. 52 (https://zenodo.org/records/16218570), one of which is illustrated in Fig. 5b. These matrices are required to compute clustering coefficients, betweenness centrality, and regional degree, with Fig. 5c displaying clustering coefficients for two arbitrarily selected participants. Supplementary analyses are also supported by available data. The frequencies underlying the analysis of participants exposed to either the smallest or largest ring as the CS+ are provided. In addition, raw behavioral data and code required to replicate the fear induction analysis shown in Supplementary Fig. 1 are included in ref. 52 (https://zenodo.org/records/16218570).

### Code availability

The analyses of behavioral data were conducted with in-house code utilizing standard functions included in Matlab (2021b). The analyses of fMRI data were conducted with in-house code also written in Matlab or its statistics and machine learning toolbox respectively. Parts of this code make use of SPM12 routines for basic neuroimaging file (i.e., read and write) operations. All in-house routines used for behavioral and fMRI data analyses are available online via the Zenodo data and code repository ([52]; https://zenodo.org/records/16218570). Further data processing steps were conducted with freely available neuroimaging software packages. First, preprocessing of structural and functional MRI scans and parts of the preprocessing of the DWI scans were conducted with SPM12 and FSL (6.0.7.17; https://fsl.fmrib.ox.ac.uk/fsl/docs/#/). Other parts of DWI scan preprocessing were conducted with Mrtrix3 (https://www.mrtrix.org/). The graph-based analyses of the brain's structural connectome were conducted with the Brain Connectivity Toolbox (https://sites.google.com/site/bctnet/; release 3rd of March 2019) for MATLAB. These freely available software packages can be received from their respective websites.

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

## Acknowledgements

We thank our participants for taking part in this study, as well as the Berlin Center for Advanced Neuroimaging (BCAN) for enabling the acquisition of MRI data, and Susan Pikol and Cynthia Kraut for their support in acquiring the data. The work was supported by the German Research Foundation (WE 5967/2-1 and WE 5967/2-2 to MW and Exc 257 to FP). LMA is participant in

the BIH Charité (Junior) (Digital) Clinician Scientist Program funded by the Charité—Universitätsmedizin Berlin, and the Berlin Institute of Health at Charité (BIH). Our funding sources did not influence the study design, the collection, analysis and interpretation of data, the writing of the report or the decision to submit the article for publication.

## Author contributions

Conceptualization, M.W.; methodology, M.W.; data acquisition, L.M.-A., R.R., and M.W.; writing, L.M.-A. and M.W.; intellectual contribution and manuscript editing, L.M.-A., R.R., J.B.-S., T.S.-H., L.M., S.F., M.S., S.M.G., S.H., and F.P.; funding acquisition and supervision, F.P. and M.W.

## Funding

## Competing interests

The authors declare no competing interests.

## Additional information

[1]Max Delbrück Center for Molecular Medicine in the Helmholtz Association, Berlin, Germany. [2]Experimental and Clinical Research Center, a cooperation between the Max Delbrück Center for Molecular Medicine in the Helmholtz Association and Charité—Universitätsmedizin Berlin, Berlin, Germany. [3]NeuroCure Clinical Research Center, Charité—Universitätsmedizin Berlin, corporate member of Freie Universität Berlin, Humboldt-Universität zu Berlin, and Berlin Institute of Health, Berlin, Germany. [4]Department of Neurology and Experimental Neurology, Charité—Universitätsmedizin Berlin, corporate member of Freie Universität Berlin, Humboldt-Universität zu Berlin, and Berlin Institute of Health, Berlin, Germany. [5]BIH Charité (Junior) (Digital) Clinician Scientist Program, Berlin Institute of Health at Charité—Universitätsmedizin Berlin, BIH Biomedical Innovation Academy, Berlin, Germany. [6]Institute for Immunology, Charité—Universitätsmedizin Berlin Campus Virchow-Klinikum (CVK), Berlin, Germany. [7]German Centre for Cardiovascular Research, (DZHK), Berlin, Germany. [8]European Molecular Biology Laboratory, Structural and Computational Biology Unit, Heidelberg, Germany. [9]Department of Neuroradiology, Charité - Universitätsmedizin Berlin, Corporate Member of Freie Universität Berlin, Humboldt-Universität zu Berlin, Berlin, Germany. [10]Department of Psychiatry and Psychotherapy, Charité—Universitätsmedizin Berlin, corporate member of Freie Universität Berlin, Humboldt-Universität zu Berlin, and Berlin Institute of Health, Berlin, Germany. [11]Department of Psychosomatic Medicine, Charité—Universitätsmedizin Berlin, corporate member of Freie Universität Berlin, Humboldt-Universität zu Berlin, and Berlin Institute of Health, Berlin, Germany. [12]Institute of Neuroimmunology and Multiple Sclerosis (INIMS), Center for Molecular Neurobiology Hamburg, Universitätsklinikum Hamburg-Eppendorf, Hamburg, Germany. [13]Berlin Center for Advanced Neuroimaging (BCAN), Charité—Universitätsmedizin Berlin, Corporate Member of Freie Universität Berlin and Humboldt-Universität zu Berlin, Berlin, Germany. ✉e-mail: martin.weygandt@mdc-berlin.de

