## [Transparent Peer Review file · Communications Medicine]

Neurobehavioral mechanisms of fear and anxiety in multiple sclerosis

Corresponding Author: Dr Martin Weygandt

Version 0:

Reviewer comments:

Reviewer #1

(Remarks to the Author)

This work explores the mechanisms underlying anxiety in people with multiple sclerosis (PwMS), with a particular focus on fear generalization. By using fMRI-based fear generalization tasks, diffusion-weighted imaging (DWI) for structural connectome analysis, and machine learning models, the researchers found that PwMS with anxiety (PwMSA) had significant fear overgeneralization compared to matched PwMS without anxiety (PwMSNA) and healthy controls (HPs). Additionally, the results suggest that anxiety is a passive reaction to MS progression rather than actively promoted by the disease.

The study has several strengths:

- Anxiety is underdiagnosed and overlooked in PwMS. Understanding its neurobehavioral underpinnings could have clinical implications, which makes this study timely and pertinent.
- The method is rigorous, using functional and structural MRI, clinical evaluation, and behavioral tasks, along with a machine learning approach.
- The statistical analysis accounted for covariates

The study has some limitations that need to be accounted for:

- The small sample size
- The use of scales instead of structural clinical evaluations to classify individuals
- Lack of longitudinal data that could allow exploring the evolution in fear generalization/anxiety ratings along with structural and functional MRI changes.
- RRMS vs. PMS do not have the same pathophysiological processes (inflammation vs. neurodegeneration). N=2 patients had SP MS.
- N=4 patients received antidepressants, which could bias fear processing outcomes. The number of patients receiving antidepressants tended to be higher in PwMSA (3 vs. 1).

Other comments are listed below for more details:

Introduction: [...] and there is ongoing debate about whether it is merely a reaction to MS progression or actively promoted by MS-driven degeneration of neural fear systems. What about MS-driven inflammation (proinflammatory cytokines) and demyelination?

Methods:

- Rating criteria need to be clarified.
- MRI machine reference needs to be specified.
- Sample size calculation is not mentioned. The number of patients appears unbalanced among the three groups (PwMSA, PwMSNA, and healthy controls).

Results: The significant group difference in fatigue and depression needs to be stated in the results section when referring to table 1.

Reviewer #2

(Remarks to the Author)

This is a very interesting study exploring whether anxiety is a passive reaction to or an active consequence of MS. The authors state in the introduction: “[is anxiety] merely a reaction to MS progression or actively promoted by MS-driven degeneration of neural fear systems.” To answer this question, groups of pwMS with and without anxiety and healthy controls completed a fear generalization fMRI task and DWI. A ML model trained on controls predicted behavioral fear generalization in both MS groups. Reduced integrity of fear generalization regions (clustering coefficient of left inferior temporal gyrus) was shown in the MS group with anxiety compared to the MS non-anxiety group. The authors propose that their results support anxiety as a passive reaction to MS rather than an active consequence.

Although the study is interesting, I do not follow the logic suggesting these findings to support anxiety as a passive reaction rather than an active consequence of MS. First of all, these explanations do not seem to necessarily be mutually exclusive. The presence of anxiety could be both reactive and a consequence of MS-driven neurodegeneration. The reasoning seems to be that since the model derived in healthy people achieved high accuracy in both MS groups, fear generalization does not come from MS disease-specific origins and is therefore reactive. By extension, the authors seem to propose that if anxiety in MS was “active” then everyone with MS would have anxiety. To my thinking, if we are trying to dissociate MS-driven degeneration of neural fear systems from reactive anxiety, the study design would need to isolate these variables by, e.g., introducing a time variable in some fashion: disease duration, time of anxiety emergence, or through longitudinal design. Further, perhaps lower connectivity of LITG is a premorbid structural brain difference putting individuals at higher risk for both anxiety and MS, with fear generalization emerging as the expected behavioral concomitant.

I think the study would be greatly strengthened by reframing, and either eliminating or clarifying the 'active' versus 'passive' model of anxiety in MS.

Minor points: In the Statistical analysis section “Structural brain connectivity, behavioral fear generalization, and anxiety in MS,” the authors say they tested their “hypothesis that network integrity of generic fear generalization areas...reflects behavioral generalization in MS...” It would be helpful to introduce this hypothesis earlier in the paper, and provide justification. Rather than a hypothesis-driven investigation, this seems like a replication of prior work in non-MS samples and a starting point for identifying the network to be compared between anxiety and non-anxiety MS groups.

It is not clear whether ‘information processing capacities’ refers to speed, and whether the fMRI task was in fact a speeded task. Were subjects instructed to respond as quickly as possible? Later in the analytic approach section, the authors note that information processing speed was excluded from their network models for DWI ‘due to the non-functional nature of the analysis.’ It is not clear what this means, and it is difficult to determine whether this was an appropriate methodological choice. This is a minor point, but given the small N for the described analysis (comparing 12 pwMSA to 30 pwMSNA) it is likely that every analytic choice makes a large impact on results.

Please provide justification for sample size selection.

Sincerely,
Vicky Leavitt

Reviewer #3

(Remarks to the Author)

In this study, Meyer-Arndt et al. investigated fear processing in individuals with and without multiple sclerosis (MS) using a fear generalization task. They compared behavioral and fMRI responses among three groups: persons with MS and anxiety (PwMSA), persons with MS without anxiety (PwMSNA), and healthy participants (HP). Their findings showed that PwMSA exhibited greater fear overgeneralization than both PwMSNA and HP. Overall, this is a timely and compelling study, presented in a clear and well-organized manner. I have a few comments that I hope the authors will consider:

1. The title, “Neurobehavioral signature of fear and anxiety,” overstates the findings. While the authors report group differences/similarities in behavior and brain activation, the evidence does not identify a “neurobehavioral signature.”
2. In the Introduction, the basis for the three hypotheses is unclear. It would strengthen the manuscript to review relevant literature and explain the rationale behind each hypothesis.
3. In the Methods section, under “fMRI fear generalization task,” all details of the task are deferred to the Supplement. This may hinder understanding, especially for readers unfamiliar with the paradigm. A brief description in the main text may improve readability.
4. In the “Neural substrates of behavioral fear generalization” subsection, the model was trained to predict participants’ perceived risk ratings. It is unclear how this prediction is providing insights about neural substrates of fear generalization. Please clarify it.
5. In the structural connectivity-based analysis, the authors state: “We tested our hypothesis that network integrity of generic fear generalization areas described in neuropsychiatric research (e.g., 11)...” This suggests a ROI-based approach. However, the analysis appears to be whole-brain. Please clarify the analytical strategy and its alignment with the stated hypothesis.
6. Figure 3 used a large space to show design matrices and brain maps that are not informative and are distracting. I suggest reorganizing and simplifying the figure for clarity.
7. Figure 5D, seems there is an outlier participant (the most upper right dot) that drove the correlations.
8. It would be interesting to explore whether anxiety symptoms (e.g., STAI-T scores) correlate with behavioral or

neuroimaging measures.

9. In the Discussion, the authors state: "Consequently, with regard to the debate of whether anxiety in MS reflects a passive reaction to MS progression or is actively promoted by it (7), our findings favor the first option." It is unclear how the results support this conclusion.

Version 1:

Reviewer comments:

Reviewer #1

(Remarks to the Author)

This is an interesting, timely, and clinically pertinent manuscript. The reviewer thanks the authors for their careful attention to the comments and suggestions, all of which have been adequately addressed in the revision. There are no further comments.

Reviewer #2

(Remarks to the Author)

Thank you for your thoughtful responses to my comments.

I'm left with one lingering thought, which relates to the dichotomy that still remains in the authors' conceptualization of anxiety: "underlying anxiety in MS remains poorly understood, and there is ongoing debate as to whether it is merely a reaction to MS progression or actively driven by MS-related pathology"

Does 'merely a reaction to...' mean a cognitive response to the experience of having MS or does it mean a physiological response to MS pathology? If the latter, it becomes difficult to separate from the 'actively driven by MS-related pathology' concept. I think the danger here is perpetuating a notion that thoughts cause anxiety, or thoughts are anxiety. However, we know that cognition operates on a time scale that is orders of magnitude slower than physiological stress responses, i.e., anxiety.

Also 'merely' is a rather pejorative way to suggest that if anxiety is only a reaction to MS progression it is of less concern than if it is 'actively driven by MS-related pathology' which the authors seem to imply would render it more 'real'? Or if I'm wrong, and that is not what they mean, a reader may infer that.

I really do appreciate the authors' thoughtfulness and do not need a response. I offer these ideas because I am supportive of their work, and hope to impart some key thoughts that may influence the way they think about anxiety in the important studies they continue to conduct.

Thank you,
Vicky Leavitt

Reviewer #3

(Remarks to the Author)

I appreciate the authors' responsiveness in addressing my comments.

Response to the reviewers' comments for manuscript COMMSMED-25-0570-T, entitled " Neurobehavioral signature of fear and anxiety in multiple sclerosis" by Meyer-Arndt and colleagues

We thank the reviewers for their positive and constructive comments. We found the suggestions to be very helpful and believe they have significantly improved the quality and clarity of our manuscript. Below, we provide detailed responses to each individual comment. Text segments in *italics* indicate changes made to the manuscript.

Before addressing the individual comments, we would like to highlight several revisions made independently of specific reviewer requests. These changes were necessary to accommodate the word limit imposed by *Communications Medicine* (5,000 words), while still thoroughly addressing comments the reviewers' feedback:

- We deleted the section in the Discussion that emphasized the strengths of our study, which began with: "The key strengths of our study [...] in our participants."
- We condensed the description of the MRI sequences in the Methods section of the main text, moving the full-length version to the Supplement.
- Similarly, we summarized the section on the processing of anatomical MRI scans in the main text and transferred the full-length version to the Supplement.

Reviewer #1

This work explores the mechanisms underlying anxiety in people with multiple sclerosis (PwMS), with a particular focus on fear generalization. By using fMRI-based fear generalization tasks, diffusion-weighted imaging (DWI) for structural connectome analysis, and machine learning models, the researchers found that PwMS with anxiety (PwMSA) had significant fear overgeneralization compared to matched PwMS without anxiety (PwMSNA) and healthy controls (HPs). Additionally, the results suggest that anxiety is a passive reaction to MS progression rather than actively promoted by the disease.

The study has several strengths:

- **Anxiety is underdiagnosed and overlooked in PwMS. Understanding its neurobehavioral underpinnings could have clinical implications, which makes this study timely and pertinent.**
- **The method is rigorous, using functional and structural MRI, clinical evaluation, and behavioral tasks, along with a machine learning approach.**
- **The statistical analysis accounted for covariates**

We thank the reviewer for this positive evaluation.

The study has some limitations that need to be accounted for:

Comment #1: The small sample size

We acknowledge that the sample size of our study may be considered small to moderate, when compared to clinical studies focusing on anatomical MRI or resting-state fMRI data. To address this concern, we have adjusted the following paragraph in the Discussion section (p. 17):

'Our study has some limitations. The number of participants might be considered moderate, but the 54 PwMS included here were pre-calculated based on data from one of our studies (Meyer-Arndt et al.; 47), which investigates fMRI activity differences between PwMS with and without depression during processing of affectively negative and neutral pictures. As we also measured (but did not analyze) STAI-T in (47), we could compute an effect size and thus a sample size for associations between brain activity in the affective picture task and STAI-T as a reference for this fear conditioning study. As pre-calculated, we included 54 PwMS, but this fixed number could not account for multi-paradigm data loss (i.e., insufficient rating behavior and missing DWI scans). Consequently, to the extent that an affective picture task can provide an effect/sample size for a fear conditioning task, one might argue that our study is slightly underpowered. However, first, moderate to strong effects were found across analyses and the sample size was thus not too small for supporting our hypotheses. Second, on practical grounds, our study can be considered large compared to existing MS fMRI studies using cognitive tasks: according to a recent review by Rocca et al. (48), the 44 PwMS and 23 HPs included in the crucial fMRI analyses exceed the sample sizes used in 81% of the reviewed cognitive studies.'

Comment #2: The use of scales instead of structural clinical evaluations to classify individuals

We thank the reviewer for raising this important point and agree that the use of structured clinical interviews to assess anxiety would likely have been superior to relying solely on self-report scales (i.e., STAI-T). To address this comment, we have moved a section originally presented in the Methods to the Discussion and expanded it to more clearly acknowledge this limitation. Specifically, we now state on p. 17:

'Another point is the use of STAI-T instead of, e.g., DSM criteria for ADs assessed by structured interviews, as the latter might have provided more fine-grained measures of anxiety. However, we chose STAI-T as it has been shown (i) to be highly sensitive to anxiety across ADs (49), (ii) to reflect neural fear conditioning responsivity (50), and (iii) because the heterogeneity of ADs would have complicated the acquisition of a homogeneous group of PwMSA during the project's three-year funding period.'

Comment #3: Lack of longitudinal data that could allow exploring the evolution in fear generalization/anxiety ratings along with structural and functional MRI changes.

We agree with the reviewer that the inclusion of longitudinal data would have enhanced the value of our cross-sectional design. To address this point, we have added the following statement to the Discussion on p. 17:

'Furthermore, the inclusion of only a single visit could be noted, as a longitudinal design would have allowed exploration of changes in fear generalization/anxiety ratings alongside structural and functional MRI alterations. However, the choice of a cross-sectional design was predefined by the grant module funding the project, which had a three-year duration. This limited timeframe, especially given partial overlap with the COVID pandemic, posed a risk to successful multi-visit participant acquisition.'

Comment #4: RRMS vs. PMS do not have the same pathophysiological processes (inflammation vs. neurodegeneration). N=2 patients had SP MS. N=4 patients received antidepressants, which could bias fear processing outcomes. The number of patients receiving antidepressants tended to be higher in PwMSA (3 vs. 1).

We agree with the reviewer that it is essential to consider the putative impact of covariates of no interest (CNI). To address this, we repeated all major analyses after excluding participants with a progressive disease type (SPMS) and those receiving antidepressants. The tables below summarize the results of these supplementary analyses.

Behavioral fear generalization

Pair of groups	Manuscript	No SPMS, no antidepressants
PwMSA vs. PwMSNA	t = -2.11, p = 0.020	t = -1.99, p = 0.028
PwMSA vs. HPs	t = -2.95, p = 0.003	t = -2.43, p = 0.010
PwMSNA vs. HPs	t = -1.63, p = 0.050	t = -1.70, p = 0.049

Neural substrates of behavioral fear generalization (based on patterns extracted from the brain's entire grey matter)

Group	Manuscript	No SPMS, no antidepressants
PwMSA	r = 0.59, p < 5 · 10 ⁻⁵	r = 0.54, p < 5 · 10 ⁻⁵
PwMSNA	r = 0.58, p < 5 · 10 ⁻⁵	r = 0.58, p < 5 · 10 ⁻⁵

Structural brain connectivity, behavioral fear generalization, and anxiety in MS

Effect	Manuscript	No SPMS, no antidepressants
Link between clustering coefficient and fear generalization across PwMSA and PwMSNA	Inferior temporal gyrus (L): t = 4.40, p _{FWE} = 0.012	Inferior temporal gyrus (L): t = 4.66, p _{FWE} = 0.024
	Hippocampus (L): t = 4.00, p _{FWE} = 0.043	Hippocampus (L): t = 4.39, p _{FWE} = 0.018
Difference in clustering coefficient PwMSA vs. PwMSNA	Inferior temporal gyrus (L): t = -2.44, p _{FWE} = 0.016	Inferior temporal gyrus (L): t = -2.88, p _{FWE} = 0.006

Given that both progressive MS type and the use of antidepressants were already modeled as CNI in the original manuscript analyses, these additional, more restrictive analyses – including only patients with RRMS not receiving antidepressants – yielded nearly identical effects. Therefore, to preserve readability and avoid redundancy, we decided not to include these supplementary analyses in the manuscript.

Comment #5: Introduction: [...] and there is ongoing debate about whether it is merely a reaction to MS progression or actively promoted by MS-driven degeneration of neural fear systems. What about MS-driven inflammation (proinflammatory cytokines) and demyelination?

We agree with the reviewer. In addition to focal neurodegenerative processes, MS-associated focal inflammation and subsequent demyelination - as well as a more diffuse inflammatory milieu within the CNS - could contribute to the development of anxiety in MS patients. Both focal inflammation/demyelination and neurodegeneration can affect brain regions and networks

implicated in the pathophysiology of anxiety. Moreover, widespread CNS inflammation has been linked to psychiatric comorbidities in MS. For instance, a study in RRMS patients demonstrated a correlation between CNS inflammation and anxiety symptoms (Rossi et al., 2017). Similarly, a murine model of chronic CNS inflammation showed comparable effects on affective behavior (Peruga et al., 2011). In response to the reviewer's comment, we have revised the sentence in question as follows (p. 3):

'The mechanisms underlying anxiety in MS remain poorly understood, and there is ongoing debate as to whether it is merely a reaction to MS progression or actively driven by MS-related pathology (such as degeneration of neural fear processing regions or inflammation and subsequent demyelination of anxiety-related white matter [WM] pathways; 7 - 9).'

- Rossi S, Studer V, Motta C, Polidoro S, Perugini J, Macchiarulo G, Giovannetti AM, Pareja-Gutierrez L, Calò A, Colonna I, Furlan R, Martino G, Centonze D. Neuroinflammation drives anxiety and depression in relapsing-remitting multiple sclerosis. *Neurology*. 2017 Sep 26;89(13):1338-1347. doi: 10.1212/WNL.0000000000004411.
- Peruga I, Hartwig S, Thöne J, Hovemann B, Gold R, Juckel G, Linker RA. Inflammation modulates anxiety in an animal model of multiple sclerosis. *Behav Brain Res*. 2011 Jun 20;220(1):20-9. doi: 10.1016/j.bbr.2011.01.018.

Comment #6: Methods: Rating criteria need to be clarified.

To address this comment, we have moved the corresponding section from the Supplement to the Statistical analysis section of the main text on p. 9:

'Specifically, we constrained the analysis to participants who showed variation in their ratings (i.e., did not rate the same risk for all stimuli), provided at least half of all 40 possible ratings for ring-shaped stimuli across both generalization runs, and rated a risk probability increasing from the CS- to the CS+. This resulted in a selection of 13 PwMSA, 31 PwMSNA and 23 HPs.'

Comment #7: Methods: MRI machine reference needs to be specified.

To address this point, we have included the following sentence on p. 6:

'All MR images were acquired with the same 3 Tesla whole-body tomograph (Magnetom Prisma, Siemens, Erlangen, Germany) and 64-channel head coil.'

Comment #8a: Methods: Sample size calculation is not mentioned.

Please see our reply to comment #1.

Comment #8b: The number of patients appears unbalanced among the three groups (PwMSA, PwMSNA, and healthy controls).

The reviewer is correct in noting that the number of participants varies across groups. However, this does not pose a problem for the statistical analyses applied. Specifically, each analysis of group differences was based on comparisons between independent groups of participants, which can be validly assessed using t-tests for independent samples.

The parametric version of this test, which relies on predefined t-distributions, has three key assumptions: First, independence of the observations - each participant must belong to only one group. Requirements two and three rely on the distributions of the underlying data. Specifically,

the second is that data for each group should be approximately normally distributed. This can be tested using specific statistical procedures. Finally, the third is homogeneity of variances across groups – if this assumption is violated, alternative versions such as Welch’s t-test can be used to account for unequal variances.

Notably, equal group sizes are not required. As shown in the equation for the t-statistic for independent samples (Watkins, 2016; p. 370), the test statistic adapts to different group sizes:

$$t = \frac{\bar{x} - \bar{y}}{\sqrt{\frac{s_X^2}{n_X} + \frac{s_Y^2}{n_Y}}}$$

The numerator reflects the difference in group means, while the denominator accounts for the group-specific standard deviations and sample sizes. This allows the test to remain valid even when groups are unbalanced. For further discussion, see Watkins (2016).

Importantly, while the theoretical basis of the t-test is useful for illustration, we did not calculate the t-statistics for group differences using the formula above. Instead, we implemented a linear regression approach, where we modeled group membership as a binary covariate of interest (CI) – coding one group as 1 and the other as 0 - and assessed its effect on the corresponding dependent variable (DV) in each analysis.

When no covariates of no interest (CNI) - such as the average rating time, log-transformed volume of hyperintense lesions in FLAIR, disease duration, disease type (RRMS vs. SPMS), or use of antidepressants (y/n) - are included in the model, this linear regression approach yields results identical to those obtained via the equation above. However, unlike the t-test, the regression framework allows for the inclusions of such CNI. As we adjusted for these covariates throughout our analyses, the regression-based approach was the method of choice.

Finally, statistical inference was conducted using a non-parametric permutation approach rather than relying on predefined t-distributions. The permutation method involves (i) randomly permuting the group assignment vector (the CI) many times, (ii) calculating the t-statistic for each permuted model, and (iii) determining the proportion of permuted t-statistics that are equal to or greater than the observed (unpermuted) t-statistic.

We emphasize this non-parametric procedure here to clarify that once the first assumption of the parametric t-test (independence of observations) was fulfilled – i.e., each participant appeared in only one group - the other two assumptions (normality and homogeneity of variance) were not required in our non-parametric setting, as permutation tests do not rely on distributional assumptions.

- Watkins JC (2016). An Introduction to the Science of Statistics: From Theory to Implementation, Preliminary Edition. <https://www.freetechbooks.com/joseph-c-watkins-a4471.html>

Comment #9: Results: The significant group difference in fatigue and depression needs to be stated in the results section when referring to table 1.

To address this point, we have now added the following to the Results section on ‘Clinical and demographic participant characteristics’ (p. 13):

‘In line with the high prevalence and frequent comorbidity of anxiety, depression, and fatigue in MS (40), PwMSA had higher scores on both parameters compared to the other two groups, while PwMSNA had higher scores than HPs. See Tab. 1 for details.’

Reviewer #2

This is a very interesting study exploring whether anxiety is a passive reaction to or an active consequence of MS. The authors state in the introduction: “[is anxiety] merely a reaction to MS progression or actively promoted by MS-driven degeneration of neural fear systems.” To answer this question, groups of pwMS with and without anxiety and healthy controls completed a fear generalization fMRI task and DWI. A ML model trained on controls predicted behavioral fear generalization in both MS groups. Reduced integrity of fear generalization regions (clustering coefficient of left inferior temporal gyrus) was shown in the MS group with anxiety compared to the MS non-anxiety group. The authors propose that their results support anxiety as a passive reaction to MS rather than an active consequence.

Dear Prof. Leavitt – thank you for your positive assessment.

Comment #1a: Although the study is interesting, I do not follow the logic suggesting these findings to support anxiety as a passive reaction rather than an active consequence of MS. First of all, these explanations do not seem to necessarily be mutually exclusive. The presence of anxiety could be both reactive and a consequence of MS-driven neurodegeneration.

The reviewer is, of course, correct: the presence of anxiety could be both reactive and a consequence of MS-driven neurodegeneration. Accordingly, we have made substantial changes throughout the revised manuscript, which are explained and detailed in our responses to comments #1b - d.

Comment #1b: The reasoning seems to be that since the model derived in healthy people achieved high accuracy in both MS groups, fear generalization does not come from MS disease-specific origins and is therefore reactive. By extension, the authors seem to propose that if anxiety in MS was “active” then everyone with MS would have anxiety. To my thinking, if we are trying to dissociate MS-driven degeneration of neural fear systems from reactive anxiety, the study design would need to isolate these variables by, e.g., introducing a time variable in some fashion: disease duration, time of anxiety emergence, or through longitudinal design.

We agree that incorporating time variables through statistical regression would be a suitable method for modeling MS-driven degeneration of neural fear systems or for identifying brain regions contributing to anxiety in an MS-specific manner.

For example, using a regression model (i) in a cross-sectional setting, one could define the volume of potentially anxiety-related brain regions across participants as the dependent variable (DV), with time of anxiety emergence (TAE) and disease duration as predictors. This approach could help determine which regions’ degeneration is explained by the TAE over and above what is accounted for by disease duration alone. If so, one might plausibly argue “that the corresponding regions are active drivers of anxiety in MS.”

However, this raises the question: how can we dissociate MS-driven degeneration of neural fear systems from reactive anxiety using this method?

Speculatively, one could test interaction effects using a more complex regression model (ii) that shows TAE explains variance in neurodegeneration only within a specific subgroup (i.e., an “active

anxiety group” in which anxiety is [more strongly] driven by neurodegeneration) while not doing so in another (i.e., the “reactive group” in which anxiety is less dependent on such degeneration). However, for this, group memberships must be known in advance (or ideally, one would need to estimate a “mixing proportion” of active and reactive anxiety for each participant, assuming MS-related anxiety is always a blend of both). Creating a suitable interaction regressor (TAE × group) would therefore be difficult.

One conceivable, but imperfect, solution might involve assigning MS patients who present with anxiety but show no degeneration in fear-processing regions – based on comparisons with healthy persons (HPs) - to the reactive group. These could then be compared to a broader group of MS patients with anxiety. However, this approach has a significant limitation: if one group is pre-defined as having no neurodegeneration and the other is not, the comparison cannot provide much insight into reactive anxiety – it would only inform us about MS-driven degeneration of fear circuits.

Moreover, switching to a longitudinal design would not necessarily resolve these issues. While longitudinal regression models can e.g., assess group differences (active vs. reactive) in trajectories of tissue degeneration, they still rely on a coding model for pre-defined groupings of “active” vs. “reactive”. Without a principled way to identify or model these groupings, the approach remains constrained.

More importantly, for the purposes of our study, neither cross-sectional nor longitudinal regression models of structural neuroimaging data would be well-suited to uncover the specific neurobehavioral mechanisms of anxiety in MS. This is because neither structural imaging nor variables such as TAE can capture short-term, stimulus-related processes – like trial-wise fear ratings – which have been shown to underlie anxiety in both the general population and psychiatric cohorts (e.g., Dymond et al., 2015; Lissek et al., 2012). It was precisely these dynamic processes that we aimed to study.

This brings us to the relevance of our chosen machine learning method, which aligns closely with the reviewer’s summary in comment #1b. Specifically, our multivariate pattern analysis (MVPA) cross-decoding approach - used in hundreds of studies (see e.g., the reviews of Peelen & Downing [2023] “Testing cognitive theories with multivariate pattern analysis of neuroimaging data” published in *Nature Human Behavior* or Kaplan et al. [2015] for a small overview) was designed to test for the consistency of neural representations across different contexts. As Kaplan et al. (2015, p. 1f) nicely summarize:

‘[...], there is now a growing appreciation of the power of machine learning techniques to provide evidence for similarity among neural patterns. In an MVPA experiment, a machine-learning classifier algorithm is typically trained on data from a subset of the experiment, and then tested on a held-out set of data that it has not seen before. [...] When a classifier can guess the identity of the testing trials with greater than chance accuracy, we conclude that the data contain information about the class of the stimuli, and that this information is consistent across the various subsets of data. Thus, by requiring learning transfer from training to testing datasets, MVPA constitutes a test for the consistency of information across different sets of data. This property of the test has begun to be exploited by neuroscientists who are interested in how neural patterns may be similar across different kinds of stimulus presentations, sensory modalities, and cognitive contexts. For instance, a classifier trained on data from visual presentation of objects may be asked to then classify neural patterns elicited by tactile presentations of the same objects. The success of learning transfer in such an experiment would provide direct evidence that the neural representations are similar across the two contexts.’

Although the generalization in their example occurs from one stimulus presentation type (visual) to another (tactile) within HPs, the conceptual logic is the same: if a model trained on one framework performs well in another, then the underlying neural representations must be similar.

Accordingly, in our study, we trained our fMRI fear decoding model exclusively on fMRI patterns and behavioral fear rating data from HPs. The fact that this model explained a substantial portion of fear rating variance in PwMS with anxiety (PwMSA: 35%) and without anxiety (PwMSNA: 34%) suggests that generic fear processing – since this is all the model “knows” given its training on HPs’ data – contributes meaningfully to fear generalization in MS.

However, because the model does not explain all of the variance, the reviewer’s comment #1a remains valid. This informed the way we revised the manuscript (see our response to comment #1d for details). As a result, we no longer use the terms “active” and “reactive” anxiety - except when referring to Margoni, Preziosa, Rocca, and Filippi (2023; reference 7 in the manuscript), who explicitly state in their review “Depressive symptoms, anxiety and cognitive impairment: emerging evidence in multiple sclerosis” (p. 264): “The lack of definite pathological substrates leads to consider anxiety as a reactive response following disease progression.”

In all other contexts, we simply state that generic fear processing mechanisms explain a substantial portion of the variance in fear ratings in both PwMSA and PwMSNA. We also emphasize the possibility that anxiety in MS may result from a combination of generic and disease-driven factors. By doing so, we also avoid giving the impression that we are suggesting anxiety in MS would be universal if it were an “active” process. Please see our response to comment #1d for a detailed account of the manuscript changes.

- Dymond S, Dunsmoor JE, Vervliet B, Roche B, Hermans D. Fear Generalization in Humans: Systematic Review and Implications for Anxiety Disorder Research. *Behav Ther.* 2015 Sep;46(5):561-82. doi: 10.1016/j.beth.2014.10.001.
- Kaplan JT, Man K, Greening SG. Multivariate cross-classification: applying machine learning techniques to characterize abstraction in neural representations. *Front Hum Neurosci.* 2015 Mar 25;9:151. doi: 10.3389/fnhum.2015.00151.
- Lissek S. Toward an account of clinical anxiety predicated on basic, neurally mapped mechanisms of Pavlovian fear-learning: the case for conditioned overgeneralization. *Depress Anxiety.* 2012; 29(4):257-63. doi: 10.1002/da.21922
- Margoni, M., Preziosa, P., Rocca, M.A. et al. Depressive symptoms, anxiety and cognitive impairment: emerging evidence in multiple sclerosis. *Transl Psychiatry* 13, 264 (2023). <https://doi.org/10.1038/s41398-023-02555-7>
- Peelen, M.V., Downing, P.E. Testing cognitive theories with multivariate pattern analysis of neuroimaging data. *Nat Hum Behav* 7, 1430–1441 (2023). <https://doi.org/10.1038/s41562-023-01680-z>

Comment #1c: Further, perhaps lower connectivity of LITG is a premorbid structural brain difference putting individuals at higher risk for both anxiety and MS, with fear generalization emerging as the expected behavioral concomitant.

We agree with the reviewer’s point. To address this, we have now added the following statement to the Discussion section (pp. 16-17):

‘An aspect that should be discussed is the possibility that the identified generic mechanisms - or alterations in regions known to contribute to generic fear processing - might represent risk factors for developing both anxiety and MS, with fear generalization emerging as an expected behavioral concomitant. However, given that these mechanisms and regions have been identified across many

studies analyzing HPs (e.g., 15, 16, 45), and considering the prevalence of MS (of e.g., 35 per 100,000 individuals globally in 2020 according to 46), this possibility does not challenge the notion that these mechanisms are indeed generic. Furthermore, given, for example, the strong negative link between behavioral fear generalization and clustering coefficient in hippocampus or inferior temporal gyrus found in our structural connectivity analysis – which included only PwMS across a wide range of behavioral generalization and structural connectivity levels - it appears unlikely that e.g., lower connectivity in such regions is a strong risk factor for developing MS.'

Comment #1d: I think the study would be greatly strengthened by reframing, and either eliminating or clarifying the 'active' versus 'passive' model of anxiety in MS.

To address comments #1a, b, and d, we implemented a variety of changes throughout the manuscript. First, we removed the following sentence from the Abstract:

'A critical debate exists as to whether anxiety in MS reflects a passive reaction to disease progression or arises from neurodegenerative processes.'

Next, we revised the last sentence of the Abstract to improve clarity:

'The fact that a machine learning model trained to associate fMRI fear response patterns with fear ratings in HPs could predict fear ratings from fMRI data across MS groups using an MVPA cross-decoding approach suggests that generic fear processing mechanisms substantially contribute to anxiety in MS.'

To clarify the MVPA cross-decoding approach, we expanded the Methods section (p. 10) as follows:

'Neural substrates of behavioral fear generalization

'We employed an MVPA cross-decoding ML approach (18, 35 - 37) to test our hypothesis that fear generalization recruits overlapping neural processing systems in HPs and PwMS with and without anxiety. Briefly, MVPA studies assume that patterns of neural activity reflect the structure of mental representations and use ML algorithms to determine whether these patterns vary systematically along a continuous response variable, such as fear ratings (e.g., 18). The principle behind cross-decoding is that 'if a ML algorithm trained on patterns from one context performs well when tested on patterns acquired in another, then the representations of the variable of interest are similar across both contexts' (38).'

Finally, we rewrote a sentence in the Discussion (p. 15) to improve precision and flow:

'By showing that an SVR model trained to associate fMRI patterns and fear ratings in HPs predicted fear rating in PwMSNA (explaining 34% of the variance) in an MVPA cross-decoding analysis, the study demonstrates that fear ratings in MS are largely driven by generic neural mechanisms. Moreover, the model's ability to generalize to PwMSA (35% explained variance) indicates that a common fear processing system accounts for the full range of generalization - from low levels in HPs to overgeneralization in PwMSA. These findings suggest that anxiety in MS may arise from a combination of generic and MS-specific mechanisms. Importantly, the functional nature of these processes, amenable to modulation via neuropsychiatric interventions (42), points to possible avenues for treating anxiety in MS.'

Comment #2: In the Statistical analysis section “Structural brain connectivity, behavioral fear generalization, and anxiety in MS,” the authors say they tested their “hypothesis that network integrity of generic fear generalization areas...reflects behavioral generalization in MS...” It would be helpful to introduce this hypothesis earlier in the paper, and provide justification. Rather than a hypothesis-driven investigation, this seems like a replication of prior work in non-MS samples and a starting point for identifying the network to be compared between anxiety and non-anxiety MS groups. We agree with this comment. To address it, we substantially revised the introduction to better clarify the motivation behind our study. Specifically, we highlighted the scarcity of research on anxiety mechanisms in MS, despite the clinical importance of this issue. We also emphasized the successful use of fear conditioning tasks in understanding anxiety mechanisms within the general population, and how these paradigms have been effectively applied in psychiatric cohorts.

Our motivation remained consistent regardless of the type of data analyzed - behavioral, functional or structural. We based our study on prior findings related to generic fear processing in the general population, which we hypothesized would also be relevant in MS. Therefore, we sought to investigate these mechanisms in the MS population.

To provide a clearer and more comprehensive rationale for our study, we have rewritten large parts of the introduction (p. 3):

“The mechanisms underlying anxiety in MS remain poorly understood, and there is ongoing debate as to whether it is merely a reaction to MS progression or actively driven by MS-related pathology (such as degeneration of neural fear processing regions or inflammation and subsequent demyelination of anxiety-related white matter [WM] pathways; 7 - 9). Specific components of anxiety, like altered fear processing - a key feature of ADs (10) - can be effectively studied using fear conditioning tasks, as shown in studies of anxiety in the general population and psychiatric patients (11 provides an overview); however, these have rarely been investigated in MS. In basic fear conditioning, a neutral stimulus is paired with an aversive unconditioned stimulus (“US”; e.g., an electric shock) and becomes a conditioned threat stimulus (“CS+”) that elicits fear on its own after repeated couplings. Another stimulus, never coupled with the US, becomes a safety cue (“CS-“). A more complex task with high real-world relevance is the “generalization gradient” paradigm, which demonstrates how individuals generalize fear from aversive to non-aversive stimuli. In this task, stimuli are selected from a perceptual continuum (e.g., comprising a large ring as CS+, a small ring as CS-, and rings of intermediate diameter also never coupled to the US but perceptually bridging the gap between the CS+ and the CS- as generalization stimuli; “GS”). Individuals with panic disorders (12), generalized anxiety disorder (13), and posttraumatic stress disorder (14) exhibit less steep declines in fear responses to GS that are increasingly dissimilar from the CS+ compared to HPs. This results in flatter gradients, indicating fear overgeneralization. Across neuroscience and psychiatric studies (15, 16), hippocampal functioning appears pivotal importance for fear generalization, as it mediates the comparison of stimulus representation to the CS+; if this comparison is inaccurate or biased, overgeneralization is promoted.

Thus, motivated by the importance of anxiety in MS, the scarcity of research on its mechanisms, and insights gained from fear conditioning tasks into the pathophysiology of anxiety in the general population and psychiatric cohorts, we employed a validated functional MRI (fMRI) fear generalization gradient task (17) combined with diffusion-weighted imaging (DWI) MRI to study fear generalization at behavioral, functional and structural level in MS.’

Furthermore, at the end of the Introduction, instead of (p. 4):

'Finally, we related patients' behavioral fear generalization to graph-based structural brain network parameters (e.g., 13) derived from DWI, hypothesizing that the network integrity of generic fear processing regions (14) reflects behavioral generalization in MS.'

... we now wrote:

'Finally, we related patients' behavioral fear generalization to graph-based structural brain network parameters (e.g., 19) derived from DWI, to test our hypothesis that network integrity of generic fear generalization areas reflects behavioral generalization in MS.'

Comment #3a: It is not clear whether "information processing capacities" refers to speed, and whether the fMRI task was in fact a speeded task. Were subjects instructed to respond as quickly as possible?

Yes, participants were explicitly instructed to respond as quickly as possible. This is detailed in the task description in Fig. 1:

'First, the appearance of a green arrowhead signaled the participant to rate the perceived risk of shock (ranging from "minimal" to "moderate" to "maximal") for the currently presented ring- or square-shaped stimulus, responding as fast as possible using an MRI-compatible response box.'

To highlight this point more clearly, we have now added a brief description in the main text in response to Reviewer 3's comment #3 (p. 6):

'Throughout all stages, participants were instructed to rate the perceived risk of shock as quickly as possible upon presentation of the rating cue.'

Comment #3b: Later in the analytic approach section, the authors note that information processing speed was excluded from their network models for DWI 'due to the non-functional nature of the analysis.' It is not clear what this means, and it is difficult to determine whether this was an appropriate methodological choice. This is a minor point, but given the small N for the described analysis (comparing 12 pwMSA to 30 pwMSNA) it is likely that every analytic choice makes a large impact on results.

We agree with this concern and took the following steps to address it:

- We have now included the information processing capacity marker (i.e., average rating times) as a covariate of no interest (CNI) in the analysis linking the three connectivity measures - including clustering coefficient - to patients' fear generalization marker. This analysis was conducted on all 42 patients who fulfilled the rating criteria, which was necessary for meaningful information processing capacity parameters.
- As in the original manuscript version, to maximize use of all 52 patients with available DWI scans in the analysis comparing clustering coefficient differences between PwMSA and PwMSNA, information processing capacity (available for 42 PwMS) was initially omitted as a CNI (see Fig. 5E). However, we also conducted a complementary analysis constrained to the 42 patients fulfilling the rating criteria, which included information processing capacity as a CNI.

Correspondingly, we have revised parts of the Methods section "Statistical analysis → Structural brain connectivity, behavioral fear generalization, and anxiety in MS" in the main text (p. 12) as follows:

'The three connectivity indices (regional degree, betweenness centrality, and clustering coefficient), computed separately for each region in the Neuromorphometrics brain atlas, were used as CI in linear regression analyses modeling patients' PIs. These analyses were conducted separately for each of the 142 regions and each connectivity measure across all 42 patients. CNI were as in analyses of group differences in behavioral generalization for PwMSA vs. PwMSNA. Permutation testing (20,000 permutations) was applied for two-sided significance testing, with Bonferroni correction for FWE. In a post-hoc analysis, connectivity parameters found significant above were tested for their ability to directly characterize MS-related anxiety using data from all 17 PwMSA and 35 PwMSNA with available DWI scans. Here, a dichotomous group vector served as the CI, and CNI were as in the prior analysis, except information processing speed was excluded as rating data were only available for 42 patients and we aimed to utilize the full DWI dataset of 52 PwMS. To assess the potential influence of information processing speed, we also conducted a complementary analysis constrained to the 42 patients with available rating data, including it as a CNI. Permutation testing (20,000 permutations) was used for inference, with Bonferroni correction for FWE in one-sided tests. The Supplement additionally includes analyses incorporating fatigue and depression as CNI.'

Furthermore, we have revised the corresponding Results section (p. 14) as follows:

'We found significant (positive) associations between the PIs and the clustering coefficient of the left inferior temporal gyrus ($t = 4.40$, $p_{FWE} = 0.012$, $f^2 = 0.61$), and left hippocampus ($t = 4.00$, $p_{FWE} = 0.043$, $f^2 = 0.50$). A post-hoc analysis showed that the clustering coefficient of the left inferior temporal gyrus was significantly lower in PwMSA compared to PwMSNA ($t = -2.44$, $p_{FWE} = 0.016$, $f^2 = 0.14$; Fig. 5). This effect remained significant when information processing speed was included as a CNI ($t = -2.28$, $p_{FWE} = 0.028$, $f^2 = 0.16$.'

We also updated Fig. 5 accordingly. Please note that the figure caption unchanged but is reproduced here for clarity:

Figure 5. Structural brain connectivity and behavioral fear generalization in MS. To demonstrate the outcome of the Anatomically Constrained Tractography conducted for the whole brain, (A) shows the tractography generated by Mrtrix3 for an arbitrarily selected person. (B) depicts an exemplary structural connectivity matrix characterized by the number of streamlines connecting pairs of regions that resulted from mapping the Neuromorphometrics brain atlas to that tractography. (C) shows the clustering coefficients for the regions in the atlas for two arbitrarily selected MS patients derived from such matrices. The diameter of the spheres depicts the regions' clustering coefficients for each of the two individuals. Labeled regions shown in orange correspond to areas for which a significant positive association between the regional clustering coefficient and the PI was found according to the Bonferroni-corrected significance

threshold across all patients in the analysis. (D) further illustrates these significant associations. Finally, (E) illustrates the significantly smaller clustering coefficients in PwMSA than PwMSNA in left inferior temporal gyrus found in the post-hoc analysis.'

Finally, we also revised the supplementary analysis titled "Structural brain connectivity, behavioral fear generalization, and anxiety in MS → Adjusting for CNI", which repeated the original analyses while additionally accounting for depression (BDI-II-scores) and fatigue (MFIS-scores) as CNI. As the results remained largely robust after inclusion of these additional CNI, we refer the reviewer to the Supplement for further details, in order to maintain the readability of this response letter.

Comment #4: Please provide justification for sample size selection.

To address this point, we have revised the following paragraph to the Discussion (p. 17):

'Our study has some limitations. The number of participants might be considered moderate, but the 54 PwMS included here were pre-calculated based on data from one of our studies (Meyer-Arndt et al.; 47), which investigates fMRI activity differences between PwMS with and without depression during processing of affectively negative and neutral pictures. As we also measured (but did not analyze) STAI-T in (47), we could compute an effect size and thus a sample size for associations between brain activity in the affective picture task and STAI-T as a reference for this fear conditioning study. As pre-calculated, we included 54 PwMS, but this fixed number could not account for multi-paradigm data loss (i.e., insufficient rating behavior and missing DWI scans). Consequently, to the extent that an affective picture task can provide an effect/sample size for a fear conditioning task, one might argue that our study is slightly underpowered. However, first, moderate to strong effects were found across analyses and the sample size was thus not too small for supporting our hypotheses. Second, on practical grounds, our study can be considered large compared to existing MS fMRI studies using cognitive tasks: according to a recent review by Rocca et al. (48), the 44 PwMS and 23 HPs included in the crucial fMRI analyses exceed the sample sizes used in 81% of the reviewed cognitive studies.'

With kind regards,

Lil Meyer-Arndt and Martin Weygandt

on behalf of all authors

Reviewer #3

In this study, Meyer-Arndt et al. investigated fear processing in individuals with and without multiple sclerosis (MS) using a fear generalization task. They compared behavioral and fMRI responses among three groups: persons with MS and anxiety (PwMSA), persons with MS without anxiety (PwMSNA), and healthy participants (HP). Their findings showed that PwMSA exhibited greater fear overgeneralization than both PwMSNA and HP. Overall, this is a timely and compelling study, presented in a clear and well-organized manner.

We thank the reviewer for this positive assessment of our work.

I have a few comments that I hope the authors will consider:

Comment #1: The title, “Neurobehavioral signature of fear and anxiety,” overstates the findings. While the authors report group differences/similarities in behavior and brain activation, the evidence does not identify a “neurobehavioral signature.” We agree with the reviewer and have accordingly revised the title to *‘Neurobehavioral mechanisms of fear and anxiety in multiple sclerosis’*.

Comment #2: In the Introduction, the basis for the three hypotheses is unclear. It would strengthen the manuscript to review relevant literature and explain the rationale behind each hypothesis.

We agree with the reviewer and have accordingly revised a large part of the Introduction (p. 3):

“The mechanisms underlying anxiety in MS remain poorly understood, and there is ongoing debate as to whether it is merely a reaction to MS progression or actively driven by MS-related pathology (such as degeneration of neural fear processing regions or inflammation and subsequent demyelination of anxiety-related white matter [WM] pathways; 7 - 9). Specific components of anxiety, like altered fear processing - a key feature of ADs (10) - can be effectively studied using fear conditioning tasks, as shown in studies of anxiety in the general population and psychiatric patients (11 provides an overview); however, these have rarely been investigated in MS. In basic fear conditioning, a neutral stimulus is paired with an aversive unconditioned stimulus (“US”; e.g., an electric shock) and becomes a conditioned threat stimulus (“CS+”) that elicits fear on its own after repeated couplings. Another stimulus, never coupled with the US, becomes a safety cue (“CS-“). A more complex task with high real-world relevance is the “generalization gradient” paradigm, which demonstrates how individuals generalize fear from aversive to non-aversive stimuli. In this task, stimuli are selected from a perceptual continuum (e.g., comprising a large ring as CS+, a small ring as CS-, and rings of intermediate diameter also never coupled to the US but perceptually bridging the gap between the CS+ and the CS- as generalization stimuli; “GS”). Individuals with panic disorders (12), generalized anxiety disorder (13), and posttraumatic stress disorder (14) exhibit less steep declines in fear responses to GS that are increasingly dissimilar from the CS+ compared to HPs. This results in flatter gradients, indicating fear overgeneralization. Across neuroscience and psychiatric studies (15, 16), hippocampal functioning appears pivotal importance for fear generalization, as it mediates the comparison of stimulus representation to the CS+; if this comparison is inaccurate or biased, overgeneralization is promoted.

Thus, motivated by the importance of anxiety in MS, the scarcity of research on its mechanisms, and insights gained from fear conditioning tasks into the pathophysiology of anxiety in the general population and psychiatric cohorts, we employed a validated functional MRI (fMRI) fear

generalization gradient task (17) combined with diffusion-weighted imaging (DWI) MRI to study fear generalization at behavioral, functional and structural level in MS.'

Comment #3: In the Methods section, under “fMRI fear generalization task,” all details of the task are deferred to the Supplement. This may hinder understanding, especially for readers unfamiliar with the paradigm. A brief description in the main text may improve readability.

To address this aspect, we revised the Methods section “fMRI fear generalization task” (p. 6) as follows:

'We implemented an fMRI task based on the paradigm developed by Lissek et al. (17) to study neurobehavioral fear generalization in MS. The task comprised three stages: pre-acquisition (assessing baseline responses to stimuli in a neutral, unconditioned state), acquisition (during which participants learned the association between US – and CS+, and the absence of US - CS- associations), and generalization (assessing responses to stimuli following conditioning). Throughout all stages, participants were instructed to rate the perceived risk of shock as quickly as possible upon presentation of the rating cue. Fig. 1 outlines key task features. Additional details, such as shock application and the calibration procedure used to individuals adjust shock intensity (i.e., US) prior to MRI, are provided in the Supplement.'

Comment #4: In the “Neural substrates of behavioral fear generalization” subsection, the model was trained to predict participants’ perceived risk ratings. It is unclear how this prediction is providing insights about neural substrates of fear generalization. Please clarify it.

In most task-based fMRI studies, a mass univariate General Linear Model (GLM) fMRI approach is used, in which voxel-wise activity is modeled across participants using statistical regression to test, for example, main effects of group membership or interactions between group and clinical variables, such as disease duration.

In contrast, our study employed a multivariate pattern analysis (MVPA) cross-decoding approach, which has been widely used in neuroimaging research (see reviews by Peelen & Downing [2023] “Testing cognitive theories with multivariate pattern analysis of neuroimaging data” published in *Nature Human Behavior*; Kaplan et al. [2015]).

The core principle of MVPA cross-decoding has been concisely summarized by Kaplan et al. (2015, p. 1f):

'[...], there is now a growing appreciation of the power of machine learning techniques to provide evidence for similarity among neural patterns. In an MVPA experiment, a machine-learning classifier algorithm is typically trained on data from a subset of the experiment, and then tested on a held-out set of data that it has not seen before. [...] When a classifier can guess the identity of the testing trials with greater than chance accuracy, we conclude that the data contain information about the class of the stimuli, and that this information is consistent across the various subsets of data. Thus, by requiring learning transfer from training to testing datasets, MVPA constitutes a test for the consistency of information across different sets of data. This property of the test has begun to be exploited by neuroscientists who are interested in how neural patterns may be similar across different kinds of stimulus presentations, sensory modalities, and cognitive contexts. For instance, a classifier trained on data from visual presentation of objects may be asked to then classify neural patterns elicited by tactile presentations of the same objects. The success of

learning transfer in such an experiment would provide direct evidence that the neural representations are similar across the two contexts.'

Applying this logic, our study yields the following insights into the neural substrates of fear generalization in MS:

1. Generic fear processing in MS: A machine learning model trained to associate fMRI-derived fear response patterns with fear ratings in HPs – a model necessarily reflecting generic fear processing (rating), as HPs are unaffected by MS or pathological anxiety - generalized well to patterns from PwMSNA, explaining 34% of the variance in their fear ratings. This indicates that fear ratings in MS are substantially driven by generic fear processing mechanisms.
2. Continuity across the anxiety spectrum: The same model also generalized to PwMSA, explaining 35% of the variance in their fear ratings. This suggests that a common, generic fear processing system, captured by the HP-derived model, underlies the entire spectrum of fear generalization, from normative levels in HPs to overgeneralization in PwMSA.
3. Neuroanatomical localization of fear representations: Beyond these whole-brain analyses, we also conducted a region-wise decoding using areas defined by the Neuromorphometrics brain atlas. These results reveal the spatial distribution of fear-related information across the brain and show substantial overlap with regions implicated in previous fear generalization studies in HPs and psychiatric populations, as discussed in detail in the Discussion section.

To highlight these findings, we revised the final sentence of the Abstract as follows:

'The fact that a machine learning model trained to associate fMRI fear response patterns with fear ratings in HPs could predict fear ratings from fMRI data across MS groups using an MVPA cross-decoding approach suggests that generic fear processing mechanisms substantially contribute to anxiety in MS.'

We also revised the beginning of the corresponding Methods section (p. 10) to better introduce the principle of MVPA cross-decoding:

'Neural substrates of behavioral fear generalization

'We employed an MVPA cross-decoding ML approach (18, 35 - 37) to test our hypothesis that fear generalization recruits overlapping neural processing systems in HPs and PwMS with and without anxiety. Briefly, MVPA studies assume that patterns of neural activity reflect the structure of mental representations and use ML algorithms to determine whether these patterns vary systematically along a continuous response variable, such as fear ratings (e.g., 18). The principle behind cross-decoding is that 'if a ML algorithm trained on patterns from one context performs well when tested on patterns acquired in another, then the representations of the variable of interest are similar across both contexts' (38).'

Finally, we revised a sentence in the Discussion (p. 15) to reflect our interpretation of the finding:

'By showing that an SVR model trained to associate fMRI patterns and fear ratings in HPs predicted fear rating in PwMSNA (explaining 34% of the variance) in an MVPA cross-decoding analysis, the study demonstrates that fear ratings in MS are largely driven by generic neural mechanisms. Moreover, the model's ability to generalize to PwMSA (35% explained variance) indicates that a common fear processing system accounts for the full range of generalization - from low levels in HPs to overgeneralization in PwMSA. These findings suggest that anxiety in MS may arise from a combination of generic and MS-specific mechanisms. Importantly, the functional nature of these processes, amenable to modulation via neuropsychiatric interventions (42), points to possible avenues for treating anxiety in MS.'

- Kaplan JT, Man K, Greening SG. Multivariate cross-classification: applying machine learning techniques to characterize abstraction in neural representations. *Front Hum Neurosci*. 2015 Mar 25;9:151. doi: 10.3389/fnhum.2015.00151.
- Peelen, M.V., Downing, P.E. Testing cognitive theories with multivariate pattern analysis of neuroimaging data. *Nat Hum Behav* 7, 1430–1441 (2023). <https://doi.org/10.1038/s41562-023-01680-z>

Comment #5: In the structural connectivity-based analysis, the authors state: “We tested our hypothesis that network integrity of generic fear generalization areas described in neuropsychiatric research (e.g., 11)...” This suggests a ROI-based approach. However, the analysis appears to be whole-brain. Please clarify the analytical strategy and its alignment with the stated hypothesis.

We thank the reviewer for pointing this out and agree that our description of the ROI-based approach – where structural connectivity measures were computed for individual brain regions (defined by a whole-brain atlas) based on anatomically constrained tractography - was not sufficiently precise. To clarify this, we have revised the relevant section of the Methods “Preprocessing of DWI scans, tractography, and graph-based connectivity modeling” to more accurately reflect our methodology. Specifically, on p. 9, we now write:

‘Finally, these matrices were entered into the Brain Connectivity Toolbox (32) to compute, individually for each region in the Neuromorphometrics brain atlas and each participant, three regional connectivity measures relevant across various brain disorders, including MS (e.g., 19). These measures included: regional degree (indicating the number of connections between a region and its neighboring regions), betweenness centrality (quantifying how often a region lies on the shortest paths between other region pairs), and clustering coefficient (reflecting the extent to which neighboring regions of a given region are also interconnected).’

In addition, we revised a sentence in the Methods section “Structural brain connectivity, behavioral fear generalization, and anxiety in MS” for clarity. Specifically, on (p. 12), we now write:

‘The three connectivity indices (regional degree, betweenness centrality, and clustering coefficient), computed separately for each region in the Neuromorphometrics brain atlas, were used as CI in linear regression analyses modeling patients’ PIs. These analyses were conducted separately for each of the 142 regions and each connectivity measure across all 42 patients.’

Comment #6: Figure 3 used a large space to show design matrices and brain maps that are not informative and are distracting. I suggest reorganizing and simplifying the figure for clarity.

We thank the reviewer for this valuable suggestion. To improve clarity and reduce visual complexity, we have simplified Fig. 3 by removing the schematic representations of the raw fMRI scans and design matrices. However, we retained the regression coefficient maps (i.e., labeled as b_{CSr} , b_{GS2} etc.) derived from the intra-participant fMRI activity modeling procedure. These maps represent the input patterns used for training and testing in our machine learning approach and are therefore essential to illustrating the methodological framework.

Accordingly, we also revised the figure legend, which now reads:

‘Figure 3. Predicting perceived risk of shock based on brain activity distributed across the entire GM. (A) Training phase of the SVR-based cross-decoding model using data from two randomly selected HPS and one of two available generalization runs per HPS. The model was exclusively trained on HPS’ data, using voxel-wise regression coefficient maps \mathbf{b} computed for each of the five ring stimuli (i.e., CS-, GS3, GS2, GS1, CS+) as described in the Methods section “fMRI preprocessing and brain activity modeling”. For space reasons, only three of the five maps (b_{CS-} , b_{GS2} , and b_{CS+}) are depicted here. The corresponding “Label” graphs in 3A depict the perceived risk of shock for each stimulus, modeled via logistic regression, with values indicated by orange dots. (B) Illustration of the testing procedure. (C) Prediction accuracy for the patterns of PwMSA and PwMSNA.’

Comment #7: Figure 5D, seems there is an outlier participant (the most upper right dot) that drove the correlations.

To address this comment, we tested the potential impact of outliers on the analysis referred to by the reviewer. We employed Cook’s distance, an well-established method for identifying relevant outliers in regression analyses, which avoids relying on intuition or visual inspection (Neter et al., 1996; Cook & Weisberg, 1982).

Before proceeding, we want to mention that the results of the analysis slightly changed from the initial to this revised manuscript version, as we repeated it with the additional covariate of no interest (CNI) “information processing speed” in the revised version in response to comment #3b of reviewer 2. As in the original version, left inferior temporal gyrus and left hippocampus

persisted following Bonferroni correction for multiple testing in this revised version, but not left amygdala and right middle occipital gyrus.

Specifically, to determine influential outliers, we computed Cook’s distance for each sample (D_i) for each of the two (instead of four in the original version) regression models whose results are depicted in the revised Fig. 5D. A sample’s D_i reflects the normalized change across regression coefficients of a model resulting from the deletion of this sample and thus reflects the sample’s influence on the model.

In the next step, we employed three frequently recommended techniques for computing D_i thresholds to determine which sample is an influential outlier. The first and most frequently used technique considers a sample influential if its D_i is larger than 1 (Cook & Weisberg, 1982). The second is to threshold D_i ’s by comparing them to the 50th percentile of an F-distribution with the numerator degrees of freedom being equal to the number of predictors in the model and the denominator degrees of freedom being equal to the number of samples minus the number of predictors. If D_i of a sample is larger than this 50th percentile, the sample is considered an influential outlier. The advantage of this method is to adapt outlier identification to the characteristics of a given analysis/regression model (Neter et al., 1996). The third method (e.g., recommended in the documentation of R; R core team, 2023) is to select samples as outliers when D_i is larger than 0.5. The below table summarizes the results.

Thresholding technique	ITG (L) t / n _{OL}	Hipp. (L) t / n _{OL}
Original/No exclusion	4.40 / 0	4.00 / 0
D_i > 1	4.40 / 1	4.00 / 1
D_i > F(0.5) p, n-p	4.40 / 1	4.00 / 1
D_i > 0.5	4.40 / 1	4.39 / 3

Abbrev.: Hipp., Hippocampus; ITG, inferior temporal gyrus; L, left; n, number of samples in a dataset; n_{OL}, number of influential outliers removed; p, number of predictors in a model; R, right; t, t-statistic.

The table shows that, across these three frequently used techniques for selecting influential outliers, their number as well as their impact on the results was small.

In particular, in 5 of 6 cases the single PwMS with a progressive disease type was selected as the only outlier. This sample was selected because its exclusion changed the corresponding regression coefficient from some value before exclusion to 0 afterwards – because the variable “progressive disease type” did not vary anymore at all afterwards.

The sample the reviewer is referring to (“the most upper right dot”) had no supra-threshold D_i – because its location did not sufficiently alter the slope of the regression function determined by all other samples. In other words, although this sample had a comparably high clustering coefficient and it could be considered an outlier from this perspective, it was *no influential* outlier. For this to be the case, the person would also have to have a higher or a much lower fear generalization score.

Please note that the removal of the single PwMS with a progressive disease type had no effect on the depicted 5 t-statistics due an interaction between (i) the fact that this sample was the only person with a progressive disease type and (ii) the way regression coefficients are computed - as $\mathbf{b} = (\mathbf{X}^T \cdot \mathbf{X})^{-1} \cdot \mathbf{X}^T \cdot \mathbf{y}$. In this equation, \mathbf{y} is a column vector containing the dependent variable (here: the fear generalization score across persons) and \mathbf{X} is a matrix consisting of column vectors including the predictors (i.e., the covariate of interest/clustering coefficient plus covariates of no interest) which are concatenated horizontally.

Specifically, if one predictor has only one non-zero element (as is the case here for the covariate of no interest “progressive disease type” which was zero for all persons with RRMS and one for

the single person with SPMS), this single non-zero element only affects its own regression coefficient/its element in **b** due to the underlying math; the coefficients of the other predictors in **b** are unaffected. Consequently, exclusion of this sample did not change the results of the models in the 5 cases where only this sample was excluded.

In sum, due to the robustness of results depicted in the table and the inconclusiveness of the existing literature with regard to which thresholding criterion is optimal, we did not exclude samples from this analysis.

- Cook DR, Weisberg S: Residuals and Influence in Regression, 1982., New York, Chapman & Hall, ISBN 0-412-24280-X
- Neter, J., M. H. Kutner, C. J. Nachtsheim, and W. Wasserman. Applied Linear Statistical Models. 4th ed. Chicago: Irwin, 1996.
- R core team (2023). A language and environment for statistical computing. R foundation for statistical computing. Vienna, Austria. URL <https://www.R-project.org/>

Comment #8: It would be interesting to explore whether anxiety symptoms (e.g., STAI-T scores) correlate with behavioral or neuroimaging measures.

We agree with the reviewer. To do so, we used the relation between t-statistics for group differences and corresponding correlation coefficients.

In particular, testing differences between pairs of groups in a covariate of interest (CI) is equivalent to computing the correlation between the CI across members of the two groups and a vector coding ones for the persons in one group and zeros for those in the other.

Consistently, the t-statistic reflecting group differences computed with a t-test (contrasting the CI scores of both groups) or a regression model using a group regressor (coding one for the persons in one group and zero for the other) can be converted into a Pearson correlation coefficient (or a partial Pearson correlation coefficient in case a regression model included not only the group regressor) according to the following formula:

$$r = \frac{t}{\sqrt{t^2 + n - 2}}$$

Given that the assignment of persons to the three groups (PwMSA, PwMSNA, HPs) in our study depended on (the presence/absence of MS and) their anxiety symptoms assessed via STAI-T scores (STAI-T score \geq 41: PwMSA; STAI-T score $<$ 41: PwMSNA, HP) and group membership is thus equivalent to a (dichotomous) variable reflecting (the presence or absence of) anxiety symptoms, we computed correlations between these dichotomous variables and behavioral fear generalization using this formula to address the comment in a first step.

Specifically, for $t = -2.11$ ($n = 44$) obtained for the difference of behavioral fear generalization in PwMSA vs. PwMSNA, the correlation between anxiety symptoms and the fear generalization score was $r = -0.21$ across the persons in these two groups. For PwMSA vs. HPs ($t = -2.95$, $n = 36$) it was $r = -0.45$, and for PwMSNA vs. HPs ($t = -1.63$, $n = 54$) it was $r = -0.22$.

In a second step, we computed a correlation based on the t-statistic determined for the difference in the clustering coefficient for left inferior temporal gyrus in PwMSA vs. PwMNSA computed in the post-hoc analysis in "Structural brain connectivity, behavioral fear generalization, and anxiety in MS." For the observed $t = -2.44$ and 44 samples in this comparison, $r = -0.33$ resulted.

Please note that we decided not to include these correlations in the revised manuscript for two reasons. First, in prior studies, we repeatedly experienced that reviewers favored

categorical/dichotomous representations of disease (i.e., group memberships) over continuous representations (i.e., severity scores). In other words, group differences between clinically established entities/groups appeared more attractive than associations between severity and the CI – maybe because it was considered more relevant from a clinical standpoint or better to interpret. Second, given the close link between t-statistics and correlations outlined above, providing both would be somewhat redundant.

Comment #9: In the Discussion, the authors state: “Consequently, with regard to the debate of whether anxiety in MS reflects a passive reaction to MS progression or is actively promoted by it (7), our findings favor the first option.” It is unclear how the results support this conclusion.

We admit that we overestimated the reach of our findings in this regard – for example because, as also reviewer #2 mentioned, anxiety in MS can simultaneously be a passive reaction to MS progression as well as resulting from MS-specific neurodegeneration of fear processing regions. Consequently, we formulated much more cautiously in the revised version of the manuscript.

Most importantly, we do no longer talk about “active” versus “reactive” anxiety except when referring to Margoni, Preziosa, Rocca and Filippi (2023; i.e., reference 7 in the manuscript) who explicitly mention in their review “Depressive symptoms, anxiety and cognitive impairment: emerging evidence in multiple sclerosis” (p. 264): “The lack of definite pathological substrates leads to consider anxiety as a reactive response following disease progression.” In all other cases, we refer to “generic fear processing mechanisms”, argue that MS anxiety can emerge as a consequence of generic and MS-specific mechanisms, etc. For the specific changes introduced, we would like to refer the reviewer to our reply to comment #4.

2nd reviewers' response for manuscript COMMSMED-25-0570-T, entitled " Neurobehavioral signature of fear and anxiety in multiple sclerosis" by Meyer-Arndt and colleagues

We thank the reviewer for the positive and constructive comment. We agree that the suggestions provided have helped us improve the manuscript. Please find our detailed response to the comment below. Text segments in italics reflect changes made to the manuscript.

Reviewer 2

Comment #1: I'm left with one lingering thought, which relates to the dichotomy that still remains in the authors' conceptualization of anxiety: "underlying anxiety in MS remains poorly understood, and there is ongoing debate as to whether it is merely a reaction to MS progression or actively driven by MS-related pathology".

Does 'merely a reaction to...' mean a cognitive response to the experience of having MS or does it mean a physiological response to MS pathology? If the latter, it becomes difficult to separate from the 'actively driven by MS-related pathology' concept. I think the danger here is perpetuating a notion that thoughts cause anxiety, or thoughts are anxiety. However, we know that cognition operates on a time scale that is orders of magnitude slower than physiological stress responses, i.e., anxiety.

Also 'merely' is a rather pejorative way to suggest that if anxiety is only a reaction to MS progression it is of less concern than if it is 'actively driven by MS-related pathology' which the authors seem to imply would render it more 'real'? Or if I'm wrong, and that is not what they mean, a reader may infer that.

We appreciate the reviewer's thoughtful and nuanced feedback. We fully agree that physiological processes play a significant role in anxiety, including in the context of MS. That said, cognitive processes - including thoughts - are also well-established mediators of anxiety. This is underscored by the efficacy of cognitive behavioral therapy (CBT), 'a first line, empirically supported intervention for anxiety disorders [...] designed to target maladaptive thoughts [...]' (Curtiss et al., 2021, p. 184), which would not be effective if cognitive processes were not involved. The effectiveness of CBT in treating anxiety is further supported by strong empirical evidence (e.g., Kaczurkin & Foa, 2015).

We would also like to clarify that dichotomizing the potential mediators of MS-related anxiety (e.g., "caused by neuropathology" vs. "reactive response following disease progression") was not necessarily our intention but rather reflects a common tendency among members of the scientific community to which we aimed to relate. For example, in their review "Depressive symptoms, anxiety and cognitive impairment: emerging evidence in multiple sclerosis", Margoni, Preziosa, Rocca, and Filippi (2023) state on p. 4:

'MRI findings. A few studies have investigated the association between anxiety symptoms and measures of brain structural and functional damage in MS patients with inconclusive results, reflecting the complexity of the disease. Early studies showed no correlation

between anxiety severity score and brain T2-hyperintense, T1-hypointense and gadolinium enhancing WM lesions [90–92]. Conversely, more recent evidence revealed that MS patients with fatigue and anxiety symptoms had larger caudate volumes and a thinner left parietal cortex compared to those without fatigue; another study showed that MS-related anxiety may have its neuropathological substrate in the septo-fornical area [93]. The lack of definite pathological substrates leads to consider anxiety as a reactive response following disease progression [91].’

Thus, our intention was to reflect this ongoing scientific debate rather than to reinforce a strict dichotomy or imply that one mechanism is more valid or “real” than the other. To avoid unintended implications and to directly address the reviewer’s concerns, we have revised the original sentence in the manuscript as follows:

Original: ‘The mechanisms underlying anxiety in MS remain poorly understood, and there is ongoing debate about whether it is merely a reaction to MS progression or actively promoted by MS-driven pathology (such as degeneration of neural fear processing regions or inflammation and subsequent demyelination of anxiety-related white matter [WM] pathways; 7 - 9).’

Specifically, on p. 4, we now write:

‘The mechanisms underlying anxiety in MS remain poorly understood, and there is ongoing debate as to whether it rather follows a cognitive reaction to MS progression or whether it is actively promoted by MS-driven pathology (such as degeneration of neural fear processing regions or inflammation and subsequent demyelination of anxiety-related white matter [WM] pathways; 7 - 9).’

References

Curtiss JE, Levine DS, Ander I, Baker AW. Cognitive-Behavioral Treatments for Anxiety and Stress-Related Disorders. *Focus (Am Psychiatr Publ)*. 2021 Jun;19(2):184-189. doi: 10.1176/appi.focus.20200045. Epub 2021 Jun 17.

Kaczurkin AN & Foa EB (2015) Cognitive-behavioral therapy for anxiety disorders: an update on the empirical evidence, *Dialogues in Clinical Neuroscience*, 17:3, 337-346, DOI: 10.31887/DCNS.2015.17.3/akaczurkin

Margoni, M., Preziosa, P., Rocca, M.A. et al. Depressive symptoms, anxiety and cognitive impairment: emerging evidence in multiple sclerosis. *Transl Psychiatry* 13, 264 (2023). <https://doi.org/10.1038/s41398-023-02555-7>